# Cost-efficient Collaboration Between On-device and Cloud Language Models

Avanika Narayan [* 1]   Dan Biderman [* 1 2]   Sabri Eyuboglu [* 1]   Avner May [3]
Scott Linderman [2]   James Zou [4]   Chris Ré [1]

## Abstract

We investigate an emerging setup in which a small, on-device language model (LM) with access to local data communicates with a frontier, cloud-hosted LM to solve real-world tasks involving financial, medical, and scientific reasoning over long documents. Can a local-remote collaboration reduce cloud inference costs while preserving quality? First, we consider a naïve collaboration protocol where the local and remote models simply chat back and forth. Because only the local model ingests the full context, this protocol reduces cloud costs by $30.4\times$, but recovers only 87% of the performance of the frontier model. We identify two key limitations of this protocol: the local model struggles to (1) follow multiple instructions at once and (2) reason over long contexts. Motivated by these observations, we propose MINION$\mathcal{S}$, a protocol in which the remote model decomposes the task into easier subtasks over shorter chunks of the document, that are executed locally in parallel. MINION$\mathcal{S}$ reduces costs by $5.7\times$ on average while recovering 97.9% of the remote-only performance. Our analysis reveals several key design choices that influence the trade-off between cost and performance in local-remote systems.

## 1. Introduction

Today's cloud-hosted frontier Language Models (LMs) can perform *data-intensive reasoning*: they can generate and refactor code across entire repositories and make decisions based on financial, legal, and medical documents. However, accessing these models is expensive: processing a

standard million-token code repository with OpenAI's o1 API costs $> \$15$ per query. At the same time, smaller LMs (1-8B parameters) are rapidly improving and can now run on personal computers (Ollama, llama.cpp) and smartphones (Mehta et al., 2024; Yi et al., 2024; Xu et al., 2024). Yet, today, these small, on-device LMs are used mostly for simple tasks such as tone adjustment and text completion (Gunter et al., 2024). They do not play a role in data-intensive reasoning tasks.

Inspired by the growing literature on multi-agent systems (Wang et al., 2024a; Guo et al., 2024), in this work we ask: *how can a small, on-device LM collaborate with a frontier LM in the cloud to reduce inference costs on data-intensive reasoning tasks?* In particular, we study the *communication protocols* that govern how the two LMs talk to each other, focusing on the tradeoff between cost and accuracy. To mimic realistic use cases, we study tasks that involve varying levels of reasoning over large volumes of medical, financial, and academic data (Islam et al., 2023; Adams et al., 2024; Dasigi et al., 2021).

As our first attempt, we study a simple communication protocol we call MINION: an unconstrained chat between the local and remote models. MINION reduces cloud costs by only "reading" the data locally, and communicating a compressed version of the context to the remote model. We show that while MINION achieves a $30.4\times$ reduction in remote model costs, it trails behind the remote-only baseline by 9.4 accuracy points on average (with an 8B local model; see Section 4 for details). In isolated ablations, we identify two key limitations of small LMs that hinder MINION's performance (Section 4):

- *Small LMs struggle to follow multiple instructions at once.* Large LMs often dispatch complex instructions. We find that splitting complex instructions into separate requests improves performance by 56%.

- *Small LMs get confused by long contexts*. Increasing context length from $< 1K$ to $> 65K$ decreases performance by 13% on a simple extraction task.

Motivated by these limitations, we propose MINION$\mathcal{S}$, an extension of MINION where the remote LM decomposes

---

[*]Equal contribution [1]Department of Computer Science, Stanford University [2]Department of Statistics, Stanford University [3]Together AI [4]Department of Biomedical Data Science, Stanford University. Correspondence to: Sabri Eyuboglu <eyuboglu@stanford.edu>.

*Proceedings of the $42^{nd}$ International Conference on Machine Learning*, Vancouver, Canada. PMLR 267, 2025. Copyright 2025 by the author(s).

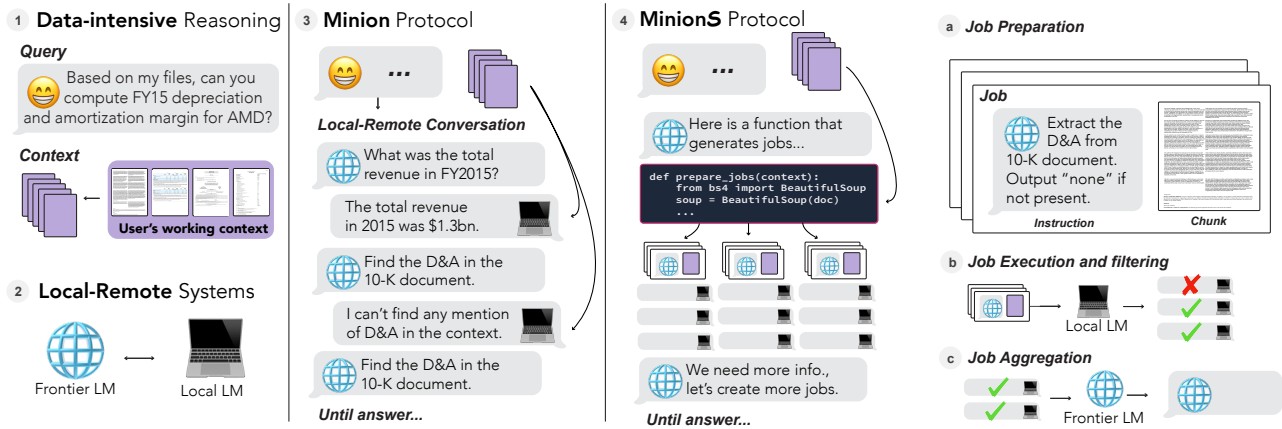

Figure 1. **Local-Remote Systems**. MINION and MINION$\mathcal{S}$ protocols. **(Left)** Problem set-up: local and remote LM collaborate on a data-intensive reasoning task. **(Center)** MINION: A simple communication protocol in which the local and remote models have an "unconstrained" back and forth chat. **(Right)** MINION$\mathcal{S}$: an extension of MINION where the remote LM decomposes a query into many *jobs* that are processed in parallel by the local model. Each job is a single instruction over a chunk of the context.

the problem into a set of *single instructions* to be performed on smaller *chunks* of the document. Crucially, the remote model has to do this *without* reading the full document, which it achieves by *generating code* that is later executed locally where the document is. More precisely, MINION$\mathcal{S}$ involves a loop over three steps:

1. **Decompose**: Given a task, the remote model writes code that decomposes it into "bite-sized" *subtasks*.

2. **Execute**: The local LM then executes the subtasks in parallel and sends a filtered selection of the responses back to the remote model.

3. **Aggregate**: The remote model aggregates the local outputs and either finalizes the solution or loops back to the Decompose step.

Averaged across tasks, MINION$\mathcal{S}$ with an 8B parameter local LM can recover 97.9% of the performance of remote-only systems at 18.0% of the cloud cost (see Figure 2). With a 3B parameter local LM, MINION$\mathcal{S}$ achieves 93.4% of the performance of remote-only systems at 16.6% of the cloud cost (see Figure 2).

We perform a detailed analysis of the design and hyperparameter space of MINION$\mathcal{S}$. Our analysis highlights several "knobs" that allow us to trade off cost for quality.

**(a) Model choice** *How does the size and family of the language models affect the cost and quality?* We show that MINION$\mathcal{S}$ became feasible only with mid-2024 releases like GPT4-TURBO and LLAMA-3.1 and is now performant with the latest 3B-parameter models running locally.

**(b) Scaling parallel workloads on-device.** *How should we*

*structure parallel workloads at the edge to maximize performance?* In Section 6.3, we study three different strategies for increasing the parallel workload on-device: (a) repeated sampling, (b) decomposition, and (c) context chunking. We show that all three can independently improve quality at the expense of increased remote cost.

**(c) Sequential communication protocols.** *Can multiple rounds of communication improve quality? At what cost?* We show that by increasing the number of sequential rounds of communication, we can pay more to improve quality.

To summarize, our main contributions are as follows:

- Propose MINION, a naïve local-remote LM communication protocol, that achieves $30.4\times$ efficiency over remote-only workloads while recovering 87% of performance.

- Propose MINION$\mathcal{S}$, an extension that overcomes the limitations we identify in MINION, achieving $5.7\times$ cost-reduction over remote-only workloads and recovering 97.9% of performance.

- Conduct an in-depth analysis of MINION$\mathcal{S}$, exploring design choices to traverse the cost-accuracy trade-off.

## 2. Related Work

*See Appendix A for an extended discussion of related work.*

This study is inspired by a large body of work that explores how to combine multiple LMs and tools to improve quality and reduce cloud inference costs. These include:

- **Retrieval-Augmented Generation (RAG)** RAG workflows append retrieved chunks to a large LM's

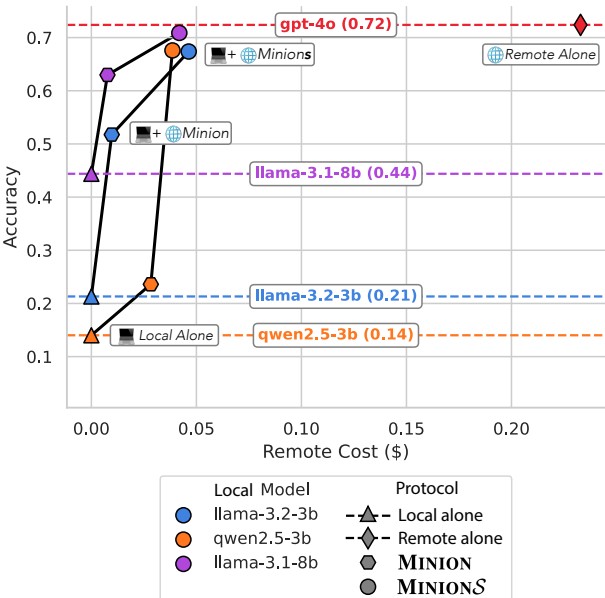

*Figure 2.* **Cost-Accuracy Tradeoff in Edge-Remote Systems.** Macro-average accuracy (*y*-axis) vs. cost (*x*-axis) across FINANCEBENCH (Islam et al., 2023), LONGHEALTH (Adams et al., 2024), and QASPER (Dasigi et al., 2021). Accuracy represents the fraction of correct predictions, while cost is the average USD per query based on GPT-4O rates (Jan 2025: $2.50/1M input tokens, $10.00/1M output tokens); see Section 3. The table compares MINION (Section 4) and MINIONS (Section 5) protocols against local-only and remote-only baselines. Points, colored by local LM, use GPT-4O as the remote LM. Exact metrics in Table 1.

prompt (Lewis et al., 2020; Karpukhin et al., 2020; Lee et al., 2019), mitigating hallucinations and externalizing knowledge. Instead of sharing raw text chunks with the remote LM, we provide the filtered output of a flexible local LM executing tasks on these chunks, further compressing the input.

- **Multi-LLM collaboration and routing** A growing body of work explores multi-agent or multi-LLM systems (Guo et al., 2024; Wang et al., 2024a) and model-routing (Chen et al., 2023; 2024a; Zheng et al., 2025). Typically, these either combine multiple large models or choose one LM from a "menu" of models for the entire task. In contrast, we explicitly study a *two-model collaboration* where the smaller local LM handles extensive on-device context, while the larger remote LM is called on selectively, reducing cloud inference costs.

- **Compound LM systems** A broader line of research integrates LMs with retrieval modules, tool use, or orchestrators (Saad-Falcon et al., 2024; Khattab et al., 2023). While they optimize accuracy or adapt prompts, they do not usually focus on asymmetric edge-cloud costs or local parallelization.

The specific techniques used in MINIONS build upon several important ideas proposed in the literature:

- **Orchestration for long-contexts** Prior works have used "divide-and-conquer" strategies to process long documents in smaller chunks (Zhang et al., 2024c; Zhou et al., 2024; Shankar et al., 2024). These methods define chunking and processing protocols, often with pipeline optimizations, but typically rely on a single large LM or symmetric LMs—unlike our asymmetric local-remote setup with cost constraints. They also don't explore parallel on-device tasks or multi-round cloud communication. Other approaches improve single-LM handling of lengthy inputs by compressing, summarizing, or streaming data. Techniques like MemGPT (Packer et al., 2023), PRISM (Jayalath et al., 2024), and writing-in-the-margins (Russak et al., 2024) store partial results or structured data across external memory. While such approaches reduce context overhead for a single LM, they do not address distributing computation across a local-remote system with distinct cost models.

- **Decomposition techniques** The decomposition techniques used in MINIONS are inspired by prior work showing how prompting for decomposition can improve small LM quality (Arora et al., 2022; Patel et al., 2022; Wu et al., 2022).The MINIONS protocol builds on using code to aid reasoning (Arora et al., 2023; Li et al., 2023).

- **Test-time sampling and verification** In MINIONS we pair repeated test-time sampling on-device with verification in the cloud. This technique is motivated by extensive literature demonstrating the promise of using test-time sampling and verification to improve reasoning capabilities (Brown et al., 2024; Song et al., 2024; Hassid et al., 2024; Snell et al., 2024; Wu et al., 2024).

- **Speculative decoding and collaborative inference.** Speculative decoding (Leviathan et al., 2023; Zhang et al., 2024a; Chen et al., 2024b) accelerates inference by *sampling* from a small "draft" model while only using a larger "target" model for verifying tokens. Collaborative inference approaches selectively sample from the target model to correct tokens identified as low-confidence ((Hao et al., 2024) places the small model on an edge device). In (Xia et al., 2023), the cloud model further retrieves external documents, generating summaries that inform the small model's generation. Our work differs by enabling the remote model to orchestrate the local model (in MINIONS, using code), extending beyond simple token evaluation or correction.

- **Prompt compression** One way to reduce cloud inference costs is to shorten input token sequences by removing tokens identified as less informative ((Jiang et al., 2023)).

Informativeness is measured by token probabilities or learned classification heads (Pan et al., 2024). In contrast, we achieve compression by offloading all raw context processing to a small LM, which can summarize and execute tasks on the context.

Some recent works have explored aspects of the local-remote setting. Several study how local-remote systems can limit leakage of private information to a cloud-hosted LM API (Siyan et al., 2024; Zhang et al., 2024b). In this work, we do not address privacy concerns, though these privacy techniques can be used in conjunction with MINION$\mathcal{S}$. Other techniques partition LM layers between local and cloud devices (Jin & Wu, 2024; Yang et al., 2024) without a multi-round dialogue. Our system is distinct in that the local LM and remote LM *collaborate in natural language* on tasks that draw on a large private context. This two-model interplay underlies our focus: reducing cloud inference costs while preserving performance.

## 3. Preliminaries

We first outline the problem setup and then provide details on how we measure accuracy and cost.

**Problem setup**  We study language tasks that involve a context $\mathbf{c}$ (*e.g.* a long document), a query $\mathbf{q}$ against that context, and a ground-truth answer $\mathbf{y}$ (see (1) in Figure 1).

> **Context (c):**  The Fiscal Year 2015 10-K report for Advanced Micro Devices, Inc.
> **Query (q):** *Compute the 2015 depreciation and amortization margin for AMD (in percentage).*
> **Answer (y):** US$394,328 million

A *local-remote* system S (see (2) in Figure 1), consists of two language models that must collaborate to solve the task—a small LM (LocalLM) running on on-device, and a large frontier LM (RemoteLM) running in the cloud. S ingests a context and a query, and applies both models in conjunction to output a predicted answer: $\hat{\mathbf{y}} \sim \mathrm{S}(\mathbf{c}, \mathbf{q})$.

**Measuring quality**  We evaluate the performance of S on a dataset $\mathcal{D} = \{(\mathbf{c}_i, \mathbf{q}_i, \mathbf{y}_i)\}_{i=1}^{N}$, via a scoring metric $s(\hat{\mathbf{y}}_i, \mathbf{y}_i)$. Here, $s(\cdot, \cdot)$ is binary (correct/incorrect) and we report *accuracy*. As baselines, we compare S to $\hat{\mathbf{y}}_{\mathrm{remote}} \sim$ RemoteLM$(\mathbf{c}, \mathbf{q})$ and $\hat{\mathbf{y}}_{\mathrm{local}} \sim$ LocalLM$(\mathbf{c}, \mathbf{q})$.

**Measuring cost**  Monetary cost (in $USD) is our primary cost metric. We assume that RemoteLM calls incur a cost while LocalLM calls are free, ignoring the fixed cost of the hardware and marginal cost of energy consumption.

More concretely, the cost of calls to RemoteLM are proportional to a weighted sum of the number of prefill (*i.e.* input) tokens and decode (*i.e.* output) tokens:

$$C_{\mathrm{remote}}(n_{\mathrm{prefill}}, n_{\mathrm{decode}}) \propto n_{\mathrm{prefill}} + \alpha \cdot n_{\mathrm{decode}},$$

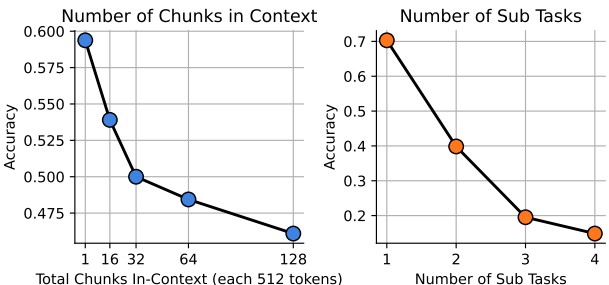

*Figure 3.* **Analysis of small LM limitations**.  Evaluation of LLAMA-3.2-3B on simple extraction tasks (see Section E.2). **(Left)** Performance drops significantly as context length increases. **(Right)** Increasing sub-task complexity reduces performance, with fewer sub-tasks yielding better results.

where $\alpha$ varies by provider ($\approx 1$– 5) (Dubey et al., 2024; Anthropic, 2024). Decode tokens are more expensive since the decoding stage is IO-bound (Leviathan et al., 2023) and has lower GPU utilization. This is because generating each decode token requires loading the full model and KV cache into GPU SRAM.

Although our focus is on monetary cost, we also measure end-to-end latency of local-remote systems. For these experiments, the LocalLM is running on a single consumer-grade GPU (*e.g.* RTX 4090, MSRP $1,599). We also include a theoretical framework for estimating latency in Appendix C.

## 4. MINION: A naïve communication protocol

In this section, we describe MINION, a baseline communication protocol, which implements a multi-turn conversation between LocalLM and RemoteLM. The former has exclusive access to the input context $\mathbf{c}$, and the latter sees only the query $\mathbf{q}$ and potential metadata about the context.

The exchange begins with system prompts: the RemoteLM is told it will collaborate with a smaller, local model that can answer context-related questions, while LocalLM is instructed to answer questions based on the provided context. The RemoteLM initiates the dialogue by asking a question and the LocalLM replies using the available context. This back-and-forth continues until RemoteLM signals it is ready to answer the query or a maximum number of steps is reached. At each turn, the RemoteLM produces a structured output with a `message` (to pass to the LocalLM) and a `decision` (to continue or terminate the conversation loop).This setup lets RemoteLM reason over long contexts without accessing them directly, using LocalLM as a retrieval and summarization interface. *See Appendix D.1 for a detailed description of the* MINION *protocol*.

We compare MINION to a baseline where RemoteLM is given the full context and the query. Excitingly, MINION

reduces RemoteLM costs by $38.13\times$, $31.3\times$, and $20.9\times$ on FINANCEBENCH, LONGHEALTH and QASPER, respectively (see Section 6.1 for dataset details). Averaged across these datasets, it closes $87.0\%$ of the quality gap between the RemoteLM versus the LocalLM operating alone.

To further close the gap, we analyze MINION conversations and find that in unconstrained chat, RemoteLM often gives LocalLM *complicated instructions* over *long contexts*. Appendix E.2 presents micro-experiments illustrating LocalLM's struggles with these instructions:

1. LocalLM **struggles to handle multiple instructions at once.** Using GPT-4O, we generate requests with varying numbers of instructions. We then show that processing each instruction separately leads to a 56 point performance improvement (see Figure 3).

2. LocalLM **struggles to reason across long contexts**. We show how increasing context length from $< 1K$ to $> 65K$ tokens can decrease performance by $13\%$ on a simple extraction instruction (see Figure 3).

Put simply, these smaller LMs are currently better equipped to answer simple queries on shorter contexts.

## 5. MINION𝒮: A decomposition-based communication protocol

Motivated by these observations, we design MINION𝒮, a simple extension of MINION that uses an iterative, divide-and-conquer approach. In each step, RemoteLM breaks the task into *jobs*—atomic instructions tied to context chunks—executed in parallel on-device (see (4) in Figure 1). Rather than a single assistant, MINION𝒮 runs a fleet handling smaller tasks. The key challenge: *How can RemoteLM create jobs for unseen context?* We solve this by having it generate code that runs locally where the context lives.

### 5.1. Protocol description

Each iteration of MINION𝒮 consists of three steps:

1. **On remote.** RemoteLM generates code to create *job specifications* for LocalLM (see 4(a) in Figure 1). The code is sent to device.

2. **On device.** The generated code is executed locally to produce the job specifications. Jobs are then executed by LocalLM, and the job outputs are filtered locally for relevance (see 4(b) in Figure 1), and sent to remote.

3. **On remote.** RemoteLM synthesizes the filtered outputs and decides whether to output an answer or continue to the next iteration (see 4(c) in Figure 1).

**Step 1: Programmatic job preparation on remote.** Our goal it to create a list of *job specifications* for LocalLM. Each job specification is a tuple including an instruction and a chunk of context to run it on $\mathbf{t}^{(i)} = (\tilde{\mathbf{q}}^{(i)}, \tilde{\mathbf{c}}^{(i)})$. The list of job specifications is denoted with $\mathbf{T} = [\mathbf{t}^{(1)}, \mathbf{t}^{(2)}, \ldots]$.

> **Instruction ($\tilde{\mathbf{q}}^{(i)}$):** Extract the total revenue for FY2015, abstain if not present. Try to look for the income statement and make sure it is from 2015.
> **Context ($\tilde{\mathbf{c}}^{(i)}$):** "Operating income for North America for the years ended..."

Crucially, to create $\mathbf{T}_t$ (at iteration $t$) without seeing the context $\mathbf{c}$, we instead have RemoteLM generate a Python function, $\mathbf{f}(\mathbf{c}, \mathbf{T}_{t-1})$, that accepts the full context $\mathbf{c}$ and previous job specifications $\mathbf{T}_{t-1}$ – both available only on-device – and outputs a new list of jobs $\mathbf{T}_t$. Access to $\mathbf{T}_{t-1}$ allows RemoteLM to use previous job outputs to allocate work to specific chunks of context.

We illustrate an abbreviated version of a Python function that was generated by RemoteLM. It includes fourteen lines of code, but can generate hundreds of jobs when executed (for a complete code example, see Appendix G).

```python
@dataclass
class Job:
    instruction: str
    chunk: str

def f(ctx: str, last_jobs: List[Job]) -> List[
    Job]:
    jobs = []
    instructions = ["Extract the total revenue
     for...", "In the statement of cash flow...
     "]
    chunks = chunk_on_pages(ctx) # chunk
     context into pages
    for chunk in chunks:
        for instr in instructions:
            for _ in range(5):
                jobs.append(Job(instr, chunk))
    return jobs
```

To generate such a function, we prompt RemoteLM with the query $\mathbf{q}$ and instruction prompt $\mathbf{p}_{\text{decompose}}$: $\mathbf{f}(\cdot, \cdot) \sim$ RemoteLM$(\mathbf{q}, \mathbf{p}_{\text{decompose}})$. The function is then executed locally to create the list of job specifications $\mathbf{T}_t$.

This strategy, which builds on work using LMs to generate code for information extraction (Arora et al., 2023; Li et al., 2023), allows us to decouple the number of unique jobs from the number tokens generated by the cloud model.

**Step 2: Job execution and filtering on-device.** In this step, we convert jobs $\mathbf{T} = [\mathbf{t}^{(1)}, \mathbf{t}^{(2)}, \ldots]$ into prompts and execute them in batch(es) locally. The prompt further guides LocalLM to produce a structured output, with fields explanation (concisely stating its reasoning), citation (a direct snippet of text supporting the answer), and answer, setting the values to null if the instruction cannot be completed confidently on this chunk. We denote each LocalLM response

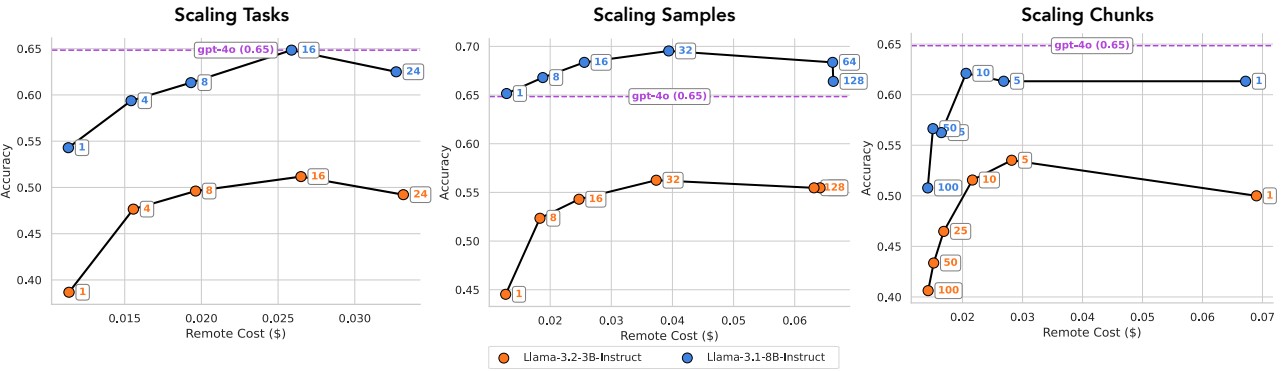

*Figure 4.* **Scaling parallel jobs on-device improves quality.** The x-axis represents tokens processed by the *remote model*, and the y-axis shows macro-average accuracy across LONGHEALTH and QASPER. The cloud model is GPT-4O. Each plot varies a different MINION$\mathcal{S}$ hyperparameter affecting parallelism, with annotated values. **(Left)** Varying the number of unique *instructions*. **(Middle)** Varying the number of unique *samples*. **(Right)** Varying the chunking granularity in code $\mathbf{f}$. See Section 5 for details.

with $\mathbf{z}^{(i)}$,

$$\mathbf{z}^{(i)} \sim \text{LocalLM}(\mathbf{t}^{(i)}, \mathbf{p}_{\text{worker}}) \qquad (1)$$

We discard any $\mathbf{z}^{(i)}$ for which the model abstained – intuitively, many instructions will be irrelevant to their paired chunks – allowing us to avoid sending unnecessary information to the RemoteLM. The surviving instruction-chunk pairs are concatenated into a single string $\mathbf{w}$.

**Step 3: Job synthesis on remote.** RemoteLM receives $\mathbf{w}$ and a synthesis prompt $\mathbf{p}_{\text{synthesize}}$, that, like in MINION, asks it to generate a structured output $\mathbf{a}$ with `decision` and `answer` fields:

$$\hat{\mathbf{y}} \sim \text{RemoteLM}(\mathbf{w}, \mathbf{p}_{\text{synthesize}}) \qquad (2)$$

If the RemoteLM decides that more information is needed, the loop continues from Step 1.

There are several ways to maintain context across MINION$\mathcal{S}$ rounds. One is to keep the entire the conversation in context, yet, this incurs significant additional cost, even with prompt caching. We experiment with two alternatives: (1) *simple retries*, in which only the RemoteLM's advice is carried over between rounds and (2) *scratchpads*, in which the RemoteLM can record what it learned from the round before proceeding to the next.

### 5.2. Protocol hyper-parameters

MINION$\mathcal{S}$ has three hyper-parameters: choice of RemoteLM and LocalLM (model choice), job preparation strategy (scale of parallel workloads on-device), and looping strategy (sequential communication protocol).

**Model choice**. Different model sizes (*e.g.* 3B vs. 8B), families (*e.g.* QWEN2.5 vs. LLAMA) can be used for both the LocalLM and the RemoteLM.

**Scale of parallel workload on-device**. MINION$\mathcal{S}$ has three

knobs for increasing the degree of task decomposition and thus, workload parallelization: (1) number of tasks per round (*i.e.* "Extract the ARR for Q1 of 2014"), (2) number of samples per task (*i.e.* number of repeated generations created with LocalLM, $\geq 1$), and (3) chunk size (*i.e.* chunk by page, chunk by paragraph, etc; smaller chunks will send more information to cloud).

**Sequential communication protocol**. In practice, it is important to cap the number of times MINION$\mathcal{S}$ can loop. After the maximum number of rounds, the synthesis prompt is modified to force the model to produce a final answer. The choice of this maximum affects accuracy and cost. The strategy for maintaining context between rounds (simple retries vs. scratchpads) is another important hyperparameter.

*We analyze these hyperparameters in Section 6.*

## 6. Results

Here, we analyze how the design of MINION$\mathcal{S}$ affects cost and quality. Our main takeaways are: **(a)** On average across three datasets, MINION$\mathcal{S}$ can recover $97.9\%$ of the performance of remote-only systems while spending $5.7\times$ less; **(b)** We identify protocol hyper-parameters that let us flexibly trade-off cost and quality; **(c)** As local models grow stronger, MINION$\mathcal{S}$ becomes increasingly cost-effective.

We structure our analysis around three core design choices:

1. **Model choice** *How does the choice of local and remote model effect cost and quality?* We examine different model types and sizes for LocalLM in Section 6.2.

2. **Scaling parallel workloads on-device** *How should we structure parallel workloads on the local device to maximize performance and minimize cost?* We highlight how scaling the local workloads can improve performance (Section 6.3) and study the effects on cost.

| Protocol | Local Model | Remote Model | Macro Avg. | | FINANCEBENCH | | LONGHEALTH | | QASPER | |
|---|---|---|---|---|---|---|---|---|---|---|
| | | | Acc. | Cost | Acc. | Cost | Acc. | Cost | Acc. | Cost |
| Remote Only | — | GPT-4O | **0.724** | $0.233 | **0.826** | $0.261 | **0.748** | $0.301 | 0.598 | $0.137 |
| Local Only | LLAMA-8B | — | 0.444 | $0.000 | 0.326 | $0.000 | 0.468 | $0.000 | 0.538 | $0.000 |
| Local Only | LLAMA-1B | — | 0.038 | $0.000 | 0.000 | $0.000 | 0.115 | $0.000 | 0.000 | $0.000 |
| Local Only | LLAMA-3B | — | 0.213 | $0.000 | 0.130 | $0.000 | 0.345 | $0.000 | 0.164 | $0.000 |
| Local Only | QWEN-3B | — | 0.140 | $0.000 | 0.087 | $0.000 | 0.177 | $0.000 | 0.156 | $0.000 |
| MINION | LLAMA-8B | GPT-4O | 0.630 | $0.008 | 0.804 | $0.007 | 0.635 | $0.010 | 0.450 | $0.007 |
| MINION | LLAMA-3B | GPT-4O | 0.518 | $0.010 | 0.698 | $0.010 | 0.482 | $0.009 | 0.372 | $0.011 |
| MINION | QWEN-3B | GPT-4O | 0.236 | $0.028 | 0.217 | $0.029 | 0.281 | $0.021 | 0.210 | $0.035 |
| MINION$\mathcal{S}$ | LLAMA-8B | GPT-4O | 0.709 | $0.042 | 0.804 | $0.053 | 0.740 | $0.054 | 0.582 | $0.019 |
| MINION$\mathcal{S}$ | LLAMA-3B | GPT-4O | 0.662 | $0.052 | 0.726 | $0.079 | 0.703 | $0.057 | 0.558 | $0.020 |
| MINION$\mathcal{S}$ | QWEN-3B | GPT-4O | 0.676 | $0.039 | 0.783 | $0.059 | 0.645 | $0.043 | **0.600** | $0.015 |

*Table 1.* **Accuracy and cost of local-remote systems.** Evaluation of cost and accuracy on 3 evaluations datasets. The table compares two edge-remote protocols—MINION (Section 4) and MINION$\mathcal{S}$ (Section 5)—against edge-only and remote-only baselines. We assess 3 local models and 1 remote model. Cost (USD) is the average per-query expense, based on GPT-4O rates (Jan 2025: $2.50M/input tokens, $10.00M/output tokens). Local model execution is assumed free (see Section 3 for cost details).

3. **Sequential communication protocol** *Can multiple rounds of communication improve quality? At what cost?* We explore this trade-off in Section 6.4.

Our findings are detailed in Sections 6.2, 6.3, and 6.4. Additionally, we analyze latency in Section 6.5.

### 6.1. Experimental setup

**Datasets and models** We evaluate MINION$\mathcal{S}$ on three benchmarks that are well suited for data-intensive reasoning: FINANCEBENCH, LONGHEALTH, and QASPER. FINANCEBENCH tests financial document understanding with complex reasoning over reports. LONGHEALTH focuses on tracking and interpreting longitudinal health records. QASPER assesses question answering over dense scientific papers. See Appendix B.0.1 for details. We use two open-source model families (LLAMA, QWEN2.5) as LocalLM and GPT-4O as RemoteLM (details in Appendix B.0.2).

### 6.2. Model choice

This section explores the model requirements and generalization capabilities of MINION$\mathcal{S}$, examining the local model sizes necessary for effective collaboration, the sensitivity of the communication protocol across different local-remote model pairings, and the longitudinal evolution of MINION$\mathcal{S}$' performance with advances in model capabilities over time.

***What size does* LocalLM *have to be in order to be effective in* MINION$\mathcal{S}$?** Our results demonstrate that MINION$\mathcal{S}$ starts being competitive with RemoteLM-only baseline at the 3B parameter model scale. When considering both the QWEN2.5 and LLAMA model families running locally, at 1B scale, MINION$\mathcal{S}$ recovers 49.5% of the GPT-4O-only baseline performance, 3B scale recovers 93.4% and 8B recovers 97.9% accuracy (see Table 1 for more details). We note that it is now feasible to run 1-8B parameter models on modern laptops (Apple M-Series) and workstations (NVIDIA DGX Spark) (Apple, 2024).

***How does the capacity of* LocalLM *affect the cost-accuracy tradeoff?*** In our system, LocalLM implicitly acts as an information encoder, optimizing the Information Bottleneck objective (Tishby et al., 2000) by compressing input context while preserving predictive information (see Appendix D.2). To measure this, we analyze the tradeoff between remote "prefill" tokens (fewer tokens indicate greater compression) and accuracy (higher accuracy means better retention). Figure 5 shows that as LocalLM size increases, representations become more compressed and accurate, improving Information Bottleneck values. Larger LocalLM models trade local FLOPs for communication, with 7–8B models being 1.53× more token-efficient than 1B models. Additionally, the QWEN2.5 family follows a different tradeoff than LLAMA, yielding more compressed representations. This suggests that as small LMs improve, local-remote systems will become increasingly cost-efficient.

***Does the performance of* MINION$\mathcal{S}$ *relative to the remote-only baseline change with increasing task complexity?*** In Tables 8 and 9, we stratify the queries by *complexity*. Our main finding is that the accuracy of MINION$\mathcal{S}$ relative to the remote-only baseline actually *improves* with increasing task complexity. For example, on simple information extraction tasks in FINANCEBENCH, MINION$\mathcal{S}$ with QWEN2.5 trails the remote only baseline by $-22.7$ accuracy points, but on tasks requiring numerical reasoning on top of extracted data, MINION$\mathcal{S}$ outperforms the remote only baseline by $+4.60$ points. On LONGHEALTH, LLAMA-8B MINION$\mathcal{S}$ trails by $-6.2$ points on questions requiring a single evidence span but outperforms the remote only baseline by $+16.0$ points on questions requiring synthesis of three evidence spans. This pattern is consistent across local model sizes.

***Is* MINION$\mathcal{S}$ *sensitive to different local/remote pairs?*** We ask whether the communication protocol in MINION$\mathcal{S}$ is invariant to changing the model types (*i.e.* LLAMA vs QWEN2.5 locally and LLAMA vs GPT-4O remotely). Our results indicate that MINION$\mathcal{S}$ performs similarly with dif-

ferent local-remote LM combinations (see the Table 1): varying the LocalLM from QWEN2.5 to LLAMA-3.2, results in performances within $\pm$ .05 performance points (see Table 1). Furthermore, we find that holding the LocalLM fixed as LLAMA-3.2-3B and varying RemoteLM from GPT-4O to LLAMA-3.3-70B leads to similar overall performances within $\pm$ 0.07 points (see Table 3 in Appendix).

***How have local / remote model capabilities changed over time, and what effects do they have on*** MINION$\mathcal{S}$***?*** In Table 4, we provide a retrospective analysis demonstrating how the quality of MINION$\mathcal{S}$ would have changed with model releases over time. From 2023 to 2025, the average performance of MINION$\mathcal{S}$ with the best models available has improved from 0.26 to 0.66 (see Table 4 in Appendix). Interestingly, it was only in July 2024 — with the release of GPT4-TURBO and LLAMA-3.1-8B — that MINION$\mathcal{S}$ could have come within 12% of the best frontier model performance at the time (see Table 4 in Appendix).

### 6.3. Scaling parallel workloads on-device

In MINION$\mathcal{S}$, there are three levers for maximizing local compute resources through parallelized, batched processing: (1) number of tasks per round, (2) number of samples taken per task, and (3) number of chunks. We ablate each, showing their impact on performance. We find that (1) and (3) are more cost effective ways of increasing performance.

***How does the number of tasks per round affect performance?*** Increasing tasks per round proxies task decomposition, with more sub-tasks enhancing decomposition. Raising tasks from 1 to 16 boosts performance by up to 14 points but doubles RemoteLM prefill costs. Optimal task count varies, but exceeding 16 reduces performance.

***How does scaling local samples affect performance?*** We explore whether increased sampling at an individual {task, context} level improves performance. Increased sampling enables us to better utilize the available compute resources while improving task-level accuracy (Brown et al., 2024). Our results indicate that increasing the number samples from 1 to 32 can improve performance on average 7.4 points, but comes at the cost of 5× the RemoteLM prefill costs. Though, increasing sampling beyond 16 starts hurting performance because the remote model struggles to parse many conflicting responses (Kuratov et al., 2024).

***What effect does chunk size have on downstream performance?*** We test whether increasing local utilization by using more chunks per task improves performance. Our results indicate that increasing # of chunks per task (by decreasing the number of "pages" per chunk from 100 to 5) leads to an 11.7 point accuracy lift. However, this lift comes with a 2.41× increase in RemoteLM prefill costs.

### 6.4. Scaling sequential communication

Both the MINION and MINION$\mathcal{S}$ communication protocols feature *sequential* communication: they allow for multiple rounds of exchange between the local and remote models.

***Does performance improve as we increase the maximum number of rounds? At what cost?*** We vary the maximum communication rounds and find it is correlated with accuracy and cost (see Figure 5). By simply increasing the maximum number of rounds in MINION from 1 to 5, we enable a 8.5-point lift in average accuracy across the three tasks (with LLAMA-3.2 on-device). However, this accuracy improvement comes at a cost: each additional round of communication increases the cost by $0.006 per query while boosting accuracy by 4.2 points.

***How should we maintain context between*** MINION$\mathcal{S}$ ***rounds?*** We experiment with two sequential protocol strategies: (1) simple retries and (2) scratchpad. See Section 5 for details of these strategies. As shown in Figure 7, both strategies show consistent increases in both accuracy and cost when increasing the maximum number of rounds, with the scratchpad strategy achieving a slightly better cost-accuracy tradeoff. Notably, each additional round of communication with the scratchpad strategy leads to a larger improvement in accuracy (6.1 accuracy points) which are mostly offset by larger increases in cost (8.6 dollars).

### 6.5. Latency of local-remote systems

Here, we compare the latency of running MINION and MINION$\mathcal{S}$ with the latency of remote-only or local-only systems.

As reported in Table 2, on a single RTX 4090, a consumer-grade graphics card for gaming (MSRP $1,599), MINION with LLAMA-3B runs only 3.5% slower than a remote only system. With LLAMA-8B, MINION's latency increases by only 44.4%. Although MINION$\mathcal{S}$ processes far more tokens than MINION or the the remote only baselines (16× more in these experiments), with careful batching and hardware-aware prefix sharing (Juravsky et al., 2024), we show we can run MINION$\mathcal{S}$ on a single RTX 4090 at 2.15× and 2.89× the latency of the remote only system.

The measurements described above are imperfect for projecting the latency that future users of local-remote systems would experience. For one, they depend on a number of unobserved variables on the remote side (*e.g.* remote hardware and user load) that are subject to change in the future. Furthermore, local hardware is improving rapidly as is the software used for serving LLMs efficiently on edge devices.

To address these limitations, in Appendix C, we include a theoretical framework for estimating the latency overhead of a local-remote system. This framework can be applied to any

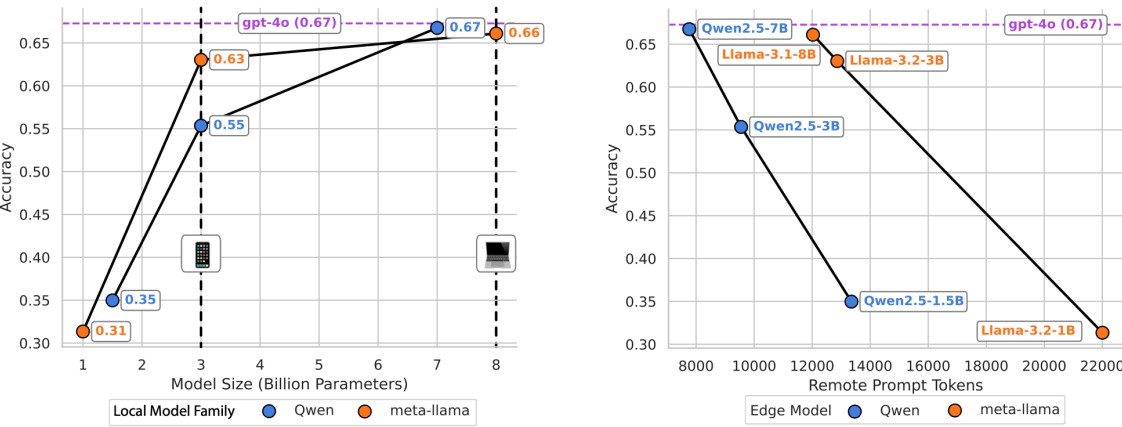

*Figure 5.* **Trade-offs in edge model performance, communication efficiency, and cost of sequential communication. (Left)** Accuracy vs. local LM size, with the purple dashed line showing the GPT-4O model baseline. **(Right)** Communication efficiency of MINION$\mathcal{S}$ with different local LMs, where larger LMs (7–8B) are more token-efficient.

| Protocol | Local | Remote | Latency (RTX 4090) | | | Latency (RTX 6000 Ada) | | |
|---|---|---|---|---|---|---|---|---|
| | | | Total | Local | Remote | Total | Local | Remote |
| Remote Only | — | GPT-4O | 11.00 | 0.00 | 11.00 | 11.00 | 0.00 | 11.00 |
| Local Only | LLAMA-8B | — | 21.13 | 21.13 | 0.00 | 19.25 | 19.25 | 0.00 |
| Local Only | LLAMA-3B | — | 11.33 | 11.33 | 0.00 | 11.01 | 11.01 | 0.00 |
| MINION | LLAMA-8B | GPT-4O | 15.88 | 9.71 | 6.17 | 29.19 | 14.16 | 15.03 |
| MINION | LLAMA-3B | GPT-4O | 11.39 | 5.51 | 5.88 | 15.32 | 5.51 | 9.81 |
| MINION$\mathcal{S}$ | LLAMA-8B | GPT-4O | 31.81 | 16.99 | 14.82 | 35.08 | 19.39 | 15.69 |
| MINION$\mathcal{S}$ | LLAMA-3B | GPT-4O | 23.65 | 9.23 | 14.42 | 26.79 | 10.76 | 16.03 |

*Table 2.* **Latency of local-remote systems.** Latency is measured in seconds on ten samples from the LONGHEALTH dataset. The LocalLM is run on a single a consumer-grade GPU: Nvidia RTX 4090 (MSRP $1,599) or Nvidia RTX 6000 Ada (∼$7,000). See Appendix C for an accompanying theoretical analysis of latency.

local and remote configuration by simply plugging in model and hardware specifications (*e.g.* memory bandwidth, model size). For instance, we estimate that running MINION$\mathcal{S}$ with LLAMA-8B on a single RTX 4090 locally and Llama 405B on a full $8\times$ H100 node remotely would incur a $4.75\times$ latency increase compared to running on the remote alone.

## 7. Discussion

MINION$\mathcal{S}$ offers insights into evolving workload distributions and the expanding role of local compute. We explore two protocols—MINION and MINION$\mathcal{S}$ — for collaboration between on-device and cloud LMs. Our results show that cloud compute costs can be reduced by $5$–$26\times$ by enabling remote LMs to delegate work to local ones. We also discuss broader implications and future directions.

**User experience** Soon, commodity hardware—laptops, smartphones, and IoT devices—will host powerful GPUs, enabling always-on local models for tasks like code refactoring, document analysis, and anomaly detection. As LocalLM capabilities grow, offloading complex, tool-augmented tasks to local compute becomes viable, reducing

our reliance on cloud API's. We take a *preliminary* step by extending the MINION framework to support local tool use (see Section E.5), showing that 3B models can complete multi-step tasks with light remote guidance—approaching GPT-4O performance while using far fewer remote tokens.

**Local-remote model co-design** There is great promise in going beyond orchestration and finetuning small models to be better minions. Our preliminary results show that MINION accuracy can be improved by simple supervised finetuning of LocalLM on the target domain (see G.2). Future work should train minions to more concisely and accurately communicate with remote models, and, eventually, jointly train two models to cooperate better. Furthermore, communication should not be limited to natural language.

**Energy efficient inference systems** Local-remote frameworks like MINION$\mathcal{S}$ can cut the energy footprint of AI by shifting compute from centralized data centers to efficient edge devices. Cloud inference is energy-intensive due to overheads like cooling and idle overprovisioning (Strubell et al., 2019; Henderson et al., 2020). Offloading to on-device models reduces reliance on power-hungry accelerators. As a first step, we quantify these savings in Section E.4, showing that MINION$\mathcal{S}$ uses up to $12\times$ less energy than fully remote execution, highlighting hybrid deployment as a promising path for sustainable AI.

**Privacy implications** MINION$\mathcal{S}$ offers a promising path to greater privacy by keeping user data on local devices during LLM inference. More broadly, local-remote systems raise important questions around privacy and security. Future work should establish formal guarantees against data leakage and secure the interface between local and remote components—for example, preventing reverse engineering or tampering of transmitted representations. Advancing adversarial robustness, secure protocols, and local model auditing will be key to hardening these systems.

## Impact Statement

MINIONS advances cost-efficient AI by demonstrating how small on-device language models can collaborate with powerful cloud-hosted models to perform data-intensive reasoning. By reducing reliance on expensive remote inference, MINIONS makes advanced AI more accessible and sustainable. This has broad societal implications, including lowering barriers to AI adoption and enhancing data privacy by keeping more computations local. However, careful consideration must be given to potential biases in small models and the security risks of local code execution.

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

# A. Extended Related Work

**Orchestration of LMs**   Recent works attempt to improve long document processing by taking a divide-and-conquer approach akin to MINION$\mathcal{S}$. Instead of using single LM calls with the entire context, the task is decomposed into smaller tasks to be executed on chunks of context. (Zhang et al., 2024c; Zhou et al., 2024) use a predefined protocol for chunk processing (defined by a prompt). (Shankar et al., 2024) performs a more involved automated pipeline optimization (via agent-based rewrite directives). Crucially, none of the works study the cost-efficient interaction between a small local LM and large remote LM and instead focus exclusively on larger LMs (70B parameters and above). Moreover, they do not explore multi-round communication patterns for document analysis.

**Long-context management techniques**   These works aim to improve *single LM* accuracy in long context tasks. (Russak et al., 2024) prefill the context using *chunks* of the document, summarize each chunk (using a predefined prompt), and aggregate the results. This improves accuracy and requires marginal additional computation. PRISM similarly (Jayalath et al., 2024) processes the context as a stream of chunks, and writes important information into a typed data structure which can be amended as needed. MemGPT (Packer et al., 2023) proposes a virtual memory paging system inspired by operating systems, where the LLM manages information across main context (akin to RAM) and external storage (akin to disk). When approaching context limits, the system actively decides what information to preserve and can later retrieve this information through paginated function calls. Orthogonally, other methods explore the usage of code for context management (Arora et al., 2023).

**Cost-efficient multi-LLM Systems**   A plethora of recent works show that multiple LMs can collaborate on a task to improve both accuracy and efficiency (Guo et al., 2024). The most similar work is perhaps (Wang et al., 2024a) which neither investigates LMs with with asymmetric capabilities nor optimizes for local compute efficiency.

**Model routing techniques**   Our work studies a collaboration of LMs, and thus should be differentiated from model routing techniques (Chen et al., 2024a; 2023) that route a prompt to the appropriate single LM that can completely answer it using the full context. This is often done for cost reduction, identifying that simple tasks can be executed by smaller and cheaper LMs.

**Compound LM systems**   Recent works explore the use of LMs as part of more elaborate pipelines that, retrieval models, tool use, and more. (Saad-Falcon et al., 2024; Khattab et al., 2023; Yuksekgonul et al., 2024) seeks to optimize the pipeline architecture and prompts using different approaches, which we do not pursue on this work.

**Retrieval-Augmented Generation (RAG)**   RAG is a hybrid approach that integrates information retrieval into the text generation process, leveraging external knowledge sources to enhance the output of language models (LMs). Instead of relying solely on parametric memory, RAG reduces the number of tokens processed by an LM by first retrieving a subset of relevant documents or document chunks and appending them as context to the LM (Lewis et al., 2020; Karpukhin et al., 2020; Lee et al., 2019; Izacard & Grave, 2021; Guu et al., 2020). This retrieval step mitigates issues such as hallucination and knowledge staleness, which are common in traditional autoregressive models (Shuster et al., 2021; Petroni et al., 2019). We differ in two ways: first, our local LM can perform tasks beyond information extraction, such as summarization or reasoning. Second, by performing arbitrary tasks on document chunks, the small LM communicates its compact answer instead of the raw document chunk, which amounts to sending fewer tokens to remote.

**Speculative decoding**   Speculative decoding (Leviathan et al., 2023; Zhang et al., 2024a; Chen et al., 2024b) techniques are addressing the different question of how to effectively sample from the distribution of a large LM by only sampling from smaller LM and using the large LM for cheaper, likelihood evaluation (using the "acceptance-complement algorithm" (Devroye, 2006)). It neither considers a collaboration between two LMs, nor attempts to minimize the communication between them.

**On-device language models for privacy**   Siyan et al. (2024); Zhang et al. (2024b) attempt to prevent leaks of private information to a cloud-hosted LM API by mediating the communication with a local privacy-aware LM that removes private information from the prompt. While the local-remote LM setup bears resemblance to ours, we do not study the aspects of privacy, but rather focus on reducing cloud costs by delegating work to devices while maintaining accuracy. Moreover, we have additional focus on local runtime efficiency.

**Local-remote systems**  Recent work has explored efficient routing patterns between local and remote computation for LM workloads, albeit without two models communicating or collaborating on a solution. (Jin & Wu, 2024) partition a single LLM with early layers on the edge and later layers in the cloud, routing to the cloud when confidence is low. (Yang et al., 2024) propose a complementary task scheduling framework that routes to cloud or local based on resource constraints and service requirements.

# B. Extended Description of Experimental Setup

## B.0.1. DATASET DETAILS

In this section we provide additional details on dataset preparation. In order to extend the context length of the problems in LONGHEALTH and QASPER, we make a few modification to the dataset.

**FINANCEBENCH**  We filter the original FINANCEBENCH to include only the numerical reasoning, resulting in a dataset of length 64. Each sample has an average context length of $142.9K(\pm79224.32)$.

**LONGHEALTH**  In the original instantiation of the LONGHEALTH dataset, each question is paired with a set of medical documents corresponding to a single patient. To increase the complexity of the dataset, we include medical documents from 10 other patients in the context. We evaluate over the entire dataset (400 problems) for results reported in Table 1. Each sample has an average context length of $120.1K(\pm1,237)$ tokens. For all ablations in Section 6, we use a fixed subset of 128 problems.

**QASPER**  Similarly, in the QASPER dataset, the original dataset provides questions that are associated with a single scientific paper. In order to increase complexity, we include 10 other papers in the context. We evaluate over a random subset of 500 problems for results reported in Table 1. Each sample has an average context length of $54281$ tokens ($\pm2403$). For all ablations in Section 6, we use a fixed subset of 128 problems.

## B.0.2. MODEL DETAILS

**Local Models**. For QWEN2.5 we use the following models: QWEN2.5-1.5-Instruct, QWEN2.5-3B-Instruct, QWEN2.5-7B-Instruct. For LLAMA, we use the following models: LLAMA-3.2-1B-Instruct, LLAMA-3.2-3B-Instruct, LLAMA-3.1-8B-Instruct.

**Remote Models**. We use GPT-4O and LLAMA-3.2-70B-Instruct, LLAMA-3.1-70B-Instruct

All "local-only" and "remote-only" experiments are run with temperature of $0.2$. For all MINION$\mathcal{S}$ experiments run in Table 1, we run the RemoteLM with a temperature of $0.0$ and LocalLM with a temperature of $0.2$ for FINANCEBENCH and $0.00001$ for QASPER and LONGHEALTH. We run our local models on A100 GPUs.

# C. Extended Discussion of Cost Model

Here, we explain in detail the costs of the different communication protocols discussed in this paper—remote-only, MINION, and MINION$\mathcal{S}$—with a strong focus on the latency of these methods. This section is organized as follows:

- Section C.1: We review background on language model inference, to motivate our cost and latency models.

- Section C.2: We present mathematical models for the latency of the remote-only, MINION, and MINION$\mathcal{S}$ protocols.

- Section C.3: We present Proposition C.1, an upper bound on the total latency of MINION$\mathcal{S}$, relative to that of the remote-only model, demonstrating that MINION$\mathcal{S}$ is not much slower than the naive approach of performing the full query in the cloud. As an example, we show that a Llama-8B model on a GTX-4090 GPU collaborating via MINION$\mathcal{S}$ with a Llama-405B model on a $8\times$H100 server is at most $4.75\times$ slower than the remote-only protocol.

## C.1. Background on language model inference

Language model inference consists of a sequence of forward passes through a model, one for prefill (*i.e.* input) followed by one for each additional token generated (*i.e.* output). At low/medium batch sizes, each forward pass after prefill is I/O bound, meaning the time it takes to load weights from memory exceeds the time it takes to actually compute the output. As the batch size increases, the computational cost of the forward pass eventually exceeds the I/O cost. Strikingly, for most

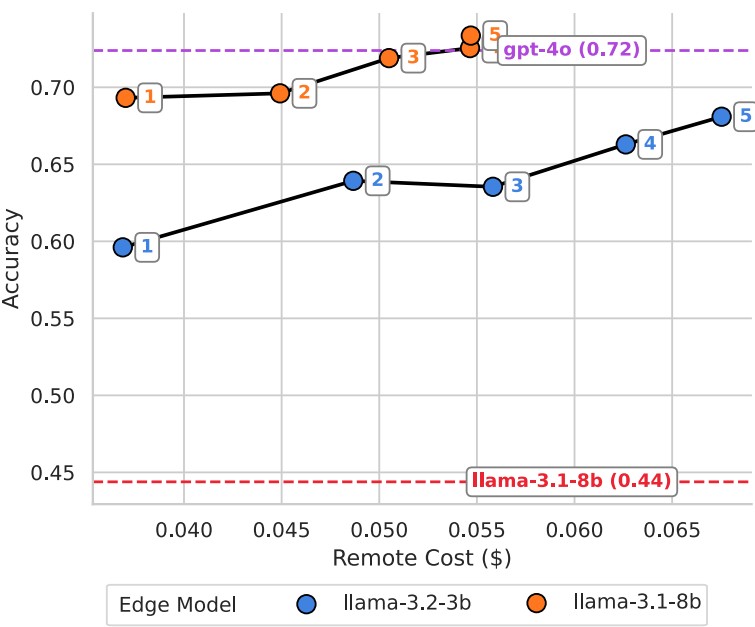

*Figure 6.* **Exploring the trade-off between cost and quality through multiple rounds.** The x-axis represents the remote model's token cost, while the y-axis shows accuracy. Point labels indicate communication rounds. The purple dashed line marks GPT-4O 's performance as a benchmark.

models and hardware, this happens at a batch size $> 100$ (Leviathan et al., 2023; Chen et al., 2024b). As a result of this transition from being I/O bound to being compute bound, we can model (as is common in the literature) the cost of running a forward pass as a piecewise linear function $C_{\mathcal{M},\mathcal{E}}(n) = \max(\lambda, \alpha \cdot n + \beta)$ of the number of tokens $n$ being processed. This is because for small $n$, the IO cost dominates (and is roughly constant as $n$ grows), whereas at larger $n$ the compute cost dominates and scales roughly linearly with $n$ (assuming $n$ is not *too* large).

In the cloud, the provider can batch generation requests from multiple users to keep hardware utilization high. Therefore, the cost of each output token is typically within a small multiple of the cost of each input token, and the total cost of processing the request scales as $n_{prefill} + \alpha \cdot n_{decode}$, for some small $\alpha \leq 5$.

On-device, we cannot assume we'll have enough concurrent user requests to form a large enough batch to achieve high utilization. As a result, the latency of a request does not scale linearly with the number of tokens. A single request can occur similar latency to hundreds run in parallel. As a result, tokens are a poor proxy for cost on-device and we instead measure latency in micro experiments (see Section 6.3).

### C.2. Latency models for all protocols: Remote-only, MINION, MINION$\mathcal{S}$

We now model the latency of each of these protocols (remote-only, MINION, MINION$\mathcal{S}$). We will then use these results in the following section to upper bound the latency of MINION$\mathcal{S}$ by a scalar multiple of the latency of the remote-only protocol.

**First, we introduce the following assumptions and notation**:

- We assume we have a local GPU (*e.g.* RTX-4090) with peak compute $F_l$ (flops/sec), and peak bandwidth $M_l$ (bytes/sec), and a remote GPU (*e.g.* H100) with peak compute $F_r$ (flops/sec), and peak bandwidth $M_r$ (bytes/sec),

- We also assume for now simple transformer architectures for both the local and remote models:
    - LocalLM: $L_l$ layers, each with $8d_l^2$ params in MLP (Up/down projections each of size $d_l \times 4d_l$, and $4d_l^2$ parameters in the $W_{Q,K,V,O}$ projections. The total memory required for the (non-embedding/LM head) parameters is thus $P_l = 2 \cdot 12L_l d_l^2$. For simplicity, we assume the memory for the LM head is small relative to $P_l$.
    - RemoteLM: Equivalent architecture to the LocalLM, but with $L_r$ layers, $d_r$ hidden dimension, and $P_r$ total non-embedding/LM-head parameter memory (again assumed to be much greater than the number of LM head

parameters).

- We model the number of input/output tokens of each protocol as follows, letting $n$ denote the number of tokens in the original document:

  - **Remote-only**: $n$ prefill tokens and $n_{out}^r$ decode tokens. Note that we assume—here and below—that the number of tokens in the query is negligible relative to $n$. We assume $n \gg n_{out}^r$ so we can effectively ignore the KV-cache load time for the output tokens.

  - **MINION**: For LocalLM, we assume $n$ prefill tokens and $n_{out}^l$ decode tokens. For RemoteLM, we assume $n_{out}^l$ prefill tokens, and $n_{out}^r$ decode tokens. In the case of multiple rounds of communication, the KV cache for the document can be stored to avoid recomputation.

  - **MINION$\mathcal{S}$**: For LocalLM, we assume $n/c$ prefill tokens per chunk ($c$ chunks total), and $n_{out}^l$ decode tokens per job (though we assume only $p$ fraction of output jobs do not abstain). For RemoteLM, we assume $J \cdot n_{out}^l \cdot p$ prefill tokens, and $n_{out}^r$ decode tokens, letting $J = cks$ denote the total number of jobs in MINION$\mathcal{S}$ ($c$ chunks, $k$ instructions, $s$ samples). In the case of multiple rounds of communication, the KV cache for each document chunk can be stored to avoid recomputation.

- Throughout, we use the fact that a $[m \times n] \cdot [n \times k]$ matmul takes $2 \cdot mnk$ flops, and assume model parameters are stored in half-precision (2 bytes/param).

We are now ready to present the latency models for the three protocols (remote-only, MINION, MINION$\mathcal{S}$).

### C.2.1. REMOTE-ONLY

- **Prefill**: We are compute bound, so time is approximately given by $total\_flops/F_r$. We can break down $total\_flops$ into the matmuls (MLP up/down projections, and QKVO operations) and attention operations.

  - **Matmuls**: $2 \cdot 12nd^2$ per layer. Equivalent to a $[n \times d_r] \cdot [d_r \times 12d_r]$ matmul.
  - **Attention**: $2 \cdot n^2 d_r$ per layer. Equivalent to $[n \times d_r] \cdot [d_r \times n]$ matmul.
  - **Time**: $L_r \cdot (24nd_r^2 + 2n^2d_r)/F_r = (nP_r + 2L_rd_rn^2)/F_r$.

- **Decode**: We are memory bound (batch size 1 for Minion), so time is approximately given by $total\_memory/M_r$ per decode step. We can break down $total\_memory$ into model parameters and KV cache.

  - **Model parameters**: $2 \cdot 12d_r^2$ bytes per layer.
  - **KV-cache**: $2 \cdot 2nd_r$ bytes per layer (K and V are each $[n \times d]$ matrices).
  - **Time**: $L_r \cdot n_{out}^r \cdot (24d_r^2 + 4nd_r)/M_r = n_{out}^r(P_r + 4L_rd_rn)/M_r$.

*Total time* is given by the sum of prefill and decode times:

$$T_{remote} = \frac{nP_r + 2L_rd_rn^2}{F_r} + \frac{n_{out}^r(P_r + 4L_rd_rn)}{M_r}$$

### C.2.2. MINION

The latency of the LocalLM in the MINION protocol can be modeled equivalently to the latency of the remote-only protocol, but replacing the remote parameters with the corresponding local ones. Thus, total local latency is:

$$T_{local}^{\text{MINION}} = \frac{nP_l + 2L_ld_ln^2}{F_l} + \frac{n_{out}^l(P_l + 4L_ld_ln)}{M_l}$$

The total remote latency can also be expressed using these same equations, but with $n_{out}^l$ prefill tokens, and $n_{out}^r$ decode tokens.

$$T_{remote}^{\text{MINION}} = \frac{n_{out}^l P_r + 2L_rd_r(n_{out}^l)^2}{F_r} + \frac{n_{out}^r(P_r + 4L_rd_rn_{out}^l)}{M_r}$$

C.2.3. MINION$\mathcal{S}$

The LocalLM latency of the MINION$\mathcal{S}$ protocol has some important differences from the MINION protocol—the prefill computation avoids cross-chunk attention (which saves time), while the decode operations can actually be compute bound if batching of the different jobs is done. We review these details below:

- **Prefill**: We are compute bound, so time is approximately given by $total\_flops/F$. We can break down $total\_flops$ into the matmuls (MLP up/down projections, and QKVO operations) and attention operations.

  - **Matmuls**: $2 \cdot 12nd_l^2$ per layer. Equivalent to $c\,[n_c \times d_l] \cdot [d_l \times 12d_l]$ matmuls (where $n_c = n/c$).
  - **Attention**: $2 \cdot cn_c^2 d_l = 2 \cdot c\,(n/c)^2 d_l = 2n^2 d_l/c$ per layer. Equivalent to $c\,[n_c \times d_l] \cdot [d_l \times n_c]$ matmuls.
  - **Time**: $L_l \cdot (24nd_l^2 + 2n^2 d_l/c)/F = (nP_l + 2L_l d_l n^2/c)/F$.

- **Decode**: We will now assume we are **compute bound during decode**, because we have many jobs ($ks$) per chunk, and many chunks ($c$) per document, which we can batch together. Thus, time is approximately given by $total\_flops/F_l$ per decode step. We can break down $total\_flops$ into matmuls and attention. The flops below are per job, per output token (so for total flops we will multiply by $n_{out}^l \cdot pcks$):

  - **Matmuls**: $2 \cdot 12d_l^2$ per layer. Equivalent to a $[1 \times d_l] \cdot [d_l \times 12d_l]$ matmul.
  - **Attention**: $2 \cdot n_c d_l = 2d_l\,n/c$ per layer. Equivalent to $[1 \times d_l] \cdot [d_l \times n_c]$ matmul.
  - **Time**: $L_l \cdot n_{out}^l \cdot pcks \cdot (24d_l^2 + 2d_l n/c)/F = n_{out}^l \cdot pcks \cdot (P_l + 2L_l d_l n/c)/F$.

The *total local latency* for MINION$\mathcal{S}$ is given by the sum of prefill and decode times: MINION$\mathcal{S}$

$$T_{local}^{\text{MINION}\mathcal{S}} = \frac{nP_l + 2L_l d_l n^2/c}{F_l} + \frac{n_{out}^l \cdot pcks \cdot (P_l + 2L_l d_l n/c)}{F_l}.$$

The *total remote latency* for MINION$\mathcal{S}$ can be expressed using the same equations as MINION, but with $pcks \cdot n_{out}^l$ prefill tokens, and $n_{out}^r$ decode tokens.

$$T_{remote}^{\text{MINION}\mathcal{S}} = \frac{(pcks \cdot n_{out}^l)P_r + 2L_r d_r (pcks \cdot n_{out}^l)^2}{F_r} + \frac{n_{out}^r (P_r + 4L_r d_r (pcks \cdot n_{out}^l))}{M_r}$$

### C.3. MINION$\mathcal{S}$ vs. remote-only comparison

**Proposition C.1.** *Assume $n_{out}^l \cdot pcks = an$, for some $a < 1$, and that $F_{r,l}$, $d_{r,l}$, and $L_{r,l}$ are all as defined in Appendix C.2. In this case, we can show that the ratio of total latency of MINION$\mathcal{S}$ vs. the remote-only protocol is upper-bounded by the following expression:*

$$\frac{T_{remote}^{\text{MINION}\mathcal{S}} + T_{local}^{\text{MINION}\mathcal{S}}}{T_{remote}} \quad < \quad 1 + \left(1 + a\right) \cdot \frac{F_r}{F_l} \cdot \frac{L_l d_l}{L_r d_r}.$$

*Proof.* Let's assume $n_{out}^l \cdot pcks = an$, for some $a < 1$.

$$
\begin{aligned}
T_{local}^{\text{MINION}\mathcal{S}} \quad &= \quad \frac{nP_l + 2L_l d_l n^2/c}{F_l} + \frac{an \cdot (P_l + 2L_l d_l n/c)}{F_l} \\[2mm]
&< \quad \left(1 + a\right) \cdot \frac{nP_l + 2L_l d_l n^2/c}{F_l} \\[2mm]
T_{remote}^{\text{MINION}\mathcal{S}} \quad &= \quad \frac{(an)P_r + 2L_r d_r (an)^2}{F_r} + \frac{n_{out}^r (P_r + 4L_r d_r (an))}{M_r} \\[2mm]
&< \quad a\left(\frac{nP_r + 2L_r d_r n^2}{F_r} + \frac{n_{out}^r 4L_r d_r n}{M_r}\right) + \frac{n_{out}^r P_r}{M_r} \\[2mm]
T_{remote} \quad &= \quad \frac{nP_r + 2L_r d_r n^2}{F_r} + \frac{n_{out}^r (P_r + 4L_r d_r n)}{M_r}
\end{aligned}
$$

Thus, it is easy to see that $\frac{T_{remote}^{\text{MINION}\mathcal{S}}}{T_{remote}} < 1$. Now let's look at $\frac{T_{local}^{\text{MINION}\mathcal{S}}}{T_{remote}}$, and show it is upper bounded by a constant:

$$
\begin{aligned}
\frac{T_{local}^{\text{MINION}\mathcal{S}}}{T_{remote}} &< \frac{\left(1+a\right) \cdot \frac{nP_l + 2L_l d_l n^2/c}{F_l}}{\frac{nP_r + 2L_r d_r n^2}{F_r}} \\
&= \left(1+a\right) \cdot \frac{F_r}{F_l} \cdot \frac{nP_l + 2L_l d_l n^2/c}{nP_r + 2L_r d_r n^2} \\
&\leq \left(1+a\right) \cdot \frac{F_r}{F_l} \cdot \max\left(\frac{P_l}{P_r}, \frac{L_l d_l}{L_r d_r c}\right) \\
&= \left(1+a\right) \cdot \frac{F_r}{F_l} \cdot \max\left(\frac{L_l d_l^2}{L_r d_r^2}, \frac{L_l d_l}{L_r d_r c}\right) \\
&< \left(1+a\right) \cdot \frac{F_r}{F_l} \cdot \frac{L_l d_l}{L_r d_r}.
\end{aligned}
$$

Thus, combining the above two results we can see that:

$$
\frac{T_{remote}^{\text{MINION}\mathcal{S}} + T_{local}^{\text{MINION}\mathcal{S}}}{T_{remote}} < 1 + \left(1+a\right) \cdot \frac{F_r}{F_l} \cdot \frac{L_l d_l}{L_r d_r}.
$$

$\square$

**Real example**: Let's assume that the local GPU is a RTX 4090 ($F_l \approx 160$ TFLOPS), the remote server is a full node of 8 H100s ($F_r \approx 8000$ TFLOPS across full node), the local model is Llama-8B ($L_l = 32$, $d_l = 4096$), and the remote model is Llama-405B ($L_l = 126$, $d_l = 16384$). Furthermore, let's assume $a \approx 0.2$, which is actually a bit larger than we see in practice. In this case:

$$
\begin{aligned}
1 + \left(1+a\right) \cdot \frac{F_r}{F_l} \cdot \frac{L_l d_l}{L_r d_r} &\approx 1 + 1.2 \cdot \frac{8000}{160} \cdot \frac{32 \cdot 4096}{126 \cdot 16384} \\
&\approx 1 + 1.2 \cdot 50 \cdot \frac{1}{16} \\
&= 4.75.
\end{aligned}
$$

Note that if we perform multiple rounds of MINION$\mathcal{S}$, this ratio gets multiplied by at most the number of rounds, though as mentioned previously, we can save time by only performing prefill on all the document chunks in the first round.

# D. Extended discussion of methods

## D.1. Extended description of MINION

In this section, we describe MINION, a baseline local-remote communication protocol. We ask whether we can reduce remote prefill tokens, and thus cost, by simply orchestrating a free-form conversation between the LocalLM and the RemoteLM in which only the LocalLM has direct access to the context $\mathbf{c}$.

The protocol proceeds with initialization step followed by a simple correspondence between the two models, which terminates when the remote model can answer the question or a maximum iteration limit is reached.

**Iteration $i = 1$: Initialize.** The RemoteLM receives the task query $\mathbf{q}$ along with a system prompt $\mathbf{p}_{\text{remote}}$ that instructs it to interact with a small LM that has full access to context. It outputs a first message $\mathbf{m}_{\text{remote}}^{(1)}$:

$$
\mathbf{m}_{\text{remote}}^{(1)} \sim \text{RemoteLM}(\mathbf{q}, \mathbf{p}_{\text{remote}})
$$

The message is then provided to LocalLM, along with the full context $\mathbf{c}$, the query $\mathbf{c}$, and a minimal system prompt $\mathbf{p}_{\text{local}}$ that instructs it to answer questions on the context:

$$\mathbf{m}_{\text{local}}^{(1)} \sim \text{LocalLM}(\mathbf{m}_{\text{remote}}^{(1)}, \mathbf{q}, \mathbf{p}_{\text{local}}, \mathbf{c})$$

**Iteration $i > 1$.** **Step 1: Message from remote to local.** RemoteLM consumes the conversation history and outputs new messages:

$$\mathbf{m}_{\text{remote}}^{(i)} \sim \text{RemoteLM}(\mathbf{m}_{\text{remote}}^{(:i-1)}, \mathbf{m}_{\text{local}}^{(:i-1)}, \mathbf{q}, \mathbf{p}_{\text{remote}})$$

In its message, RemoteLM indicates whether it has sufficient information to terminate the loop and answer the question, or alternatively raises additional questions.

**Step 2: Message from local to remote** LocalLM consumes the latest remote message and conversation history, and outputs $\mathbf{m}_{\text{local}}^{(i)}$.

$$\mathbf{m}_{\text{local}}^{(i)} \sim \text{LocalLM}(\mathbf{m}_{\text{remote}}^{(:i-1)}, \mathbf{m}_{\text{local}}^{(:i-1)}, \mathbf{q}, \mathbf{p}_{\text{local}}, \mathbf{c})$$

We then increment the iteration $i$ and loop back to **Step 1** until the break condition is met or we reach a maximum number of iterations.

### D.2. Information Bottleneck Perspective

How does local model capacity affect the cost-accuracy tradeoff?

The Information Bottleneck (IB) principle (Tishby et al., 2000) provides a useful analogy. One communication round of a local-remote system does as follows:

$$\mathbf{z} \sim p(\mathbf{z}|\mathbf{c}) \quad \text{[Extract info. from context]}$$
$$\mathbf{y} \sim p(\mathbf{y}|\mathbf{z}) \quad \text{[Predict outcome from extracted info]}$$

The IB principle seeks to find a $p(\mathbf{z} \mid \mathbf{c})$, our LocalLM, as follows:

$$\min_{p(\mathbf{z}|\mathbf{c})} \Big[ I(C;Z) \ - \ \beta \, I(Z;Y) \Big], \tag{3}$$

*i.e.* find a mapping that forces the latent representation to be maximally informative of the label $I(Z;Y)$ and minimally informative of the input $I(C;Z)$, with a tradeoff parameter $\beta$. Here, we do not optimize the mapping $p(\mathbf{z} \mid \mathbf{c})$ but instead only get to choose it by setting LocalLM.

Since we cannot compute these quantities in closed form for nonlinear distributions over tokens, we use (coarse) empirical proxies as follows. As a proxy for $I(C;Z)$, we compute the number of prefill tokens sent to RemoteLM, capturing the intuition that more tokens carry more information on the input. $I(Z;Y)$ is estimated as the average accuracy of the local-remote system, quantifying the preservation of task-relevant information in $\mathbf{z}$. While these proxies do not exactly match the underlying mutual informations, they capture the core tension of compressing the input vs. preserving predictive power.

We plot these quantities in Figure **??**. We find that across both QWEN2.5 and LLAMA model families, as we increase LocalLM size, we send fewer tokens to RemoteLM ($\approx I(C;Z) \downarrow$), and improve accuracy ($\approx I(Z;Y) \downarrow$). We find that LLAMA has higher $\approx I(C;Z)$ and higher $\approx I(Z;Y)$.

## E. Extended Results

### E.1. Model Analysis

We include additional experiment results from Section 6.2. In Table 3 we show the effects of varying RemoteLM on MINION$\mathcal{S}$. In Table 4, we show the performance of MINION$\mathcal{S}$ using the best in-class models at the time (from late 2023 to late 2024).

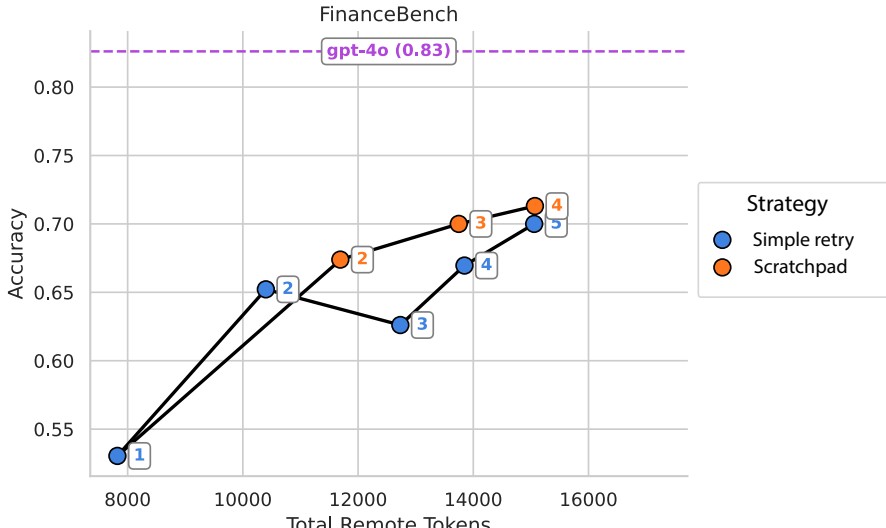

*Figure 7.* **Comparing strategies for maintaining context between MINION$\mathcal{S}$ rounds.** The x-axis represents the number of tokens processed by the *remote model*, while the y-axis shows the accuracy achieved.

| Local Model | Remote Model | Release Date | Accuracy (Longhealth) | Accuracy (QASPER) | Accuracy (Finance) |
|---|---|---|---|---|---|
| llama-3B | gpt-4o | May 2024 | 0.7025 | 0.598 | 0.7826 |
| llama-3B | gpt-4-turbo | April 2024 | 0.6247 | 0.614 | 0.6304 |
| llama-3B | gpt-3.5-turbo-0125 | Jan 2024 | 0.2157 | 0.4314 | 0.1707 |
| llama-3B | gpt4o-mini | July 2024 | 0.6275 | 0.568 | 0.6522 |
| llama-3B | llama3-70B-Instruct-Turbo | April 2024 | 0.3525 | 0.144 | 0.1818 |
| llama-3B | llama3.1-70B-Instruct-Turbo | July 2024 | 0.6193 | 0.514 | 0.4348 |
| llama-3B | llama3.3-70B-Instruct-Turbo | December 2024 | 0.6658 | 0.534 | 0.6739 |

*Table 3.* Accuracy Results for Longhealth, QASPER, and Finance across Various Models

### E.2. MINION LocalLM **Analysis**

We perform an empirical analysis evaluating the robustness of LocalLM. We perform experiments to evaluate two axes of model capabilities: (1) ability to reason over long contexts and (2) ability to solve multi-part queries. To test (1) and (2) we curate a synthetic dataset built over the FINANCEBENCH dataset wherein we use GPT-4O to construct an extraction based question-answering dataset over chunks (length 512 tokens) of documents in the FINANCEBENCH dataset. We then construct two settings evaluating over LLAMA-3.2-3B-Instruct.

**Long Context Reasoning**: To evaluate long-context reasoning, we concatenate between {1,16,32,64,128} chunks to construct the context. At least one chunk in the concatenated context contains the ground truth result. As seen in Table 5, increasing the context length from 512 to 65.5K tokens leads to a 13 point drop in accuracy.

**Multi-step Queries** To evaluate the ability of LocalLM to fulfill multi-step queries, we construct queries that have between {1,2,3,4} sub-tasks. Our results indicate increasing from 1 to sub-tasks leads to a 56.3 point drop in accuracy (see Table 6).

| Local Model | Remote Model | Accuracy (Longhealth) | Accuracy (QASPER) | System Date |
|---|---|---|---|---|
| Llama-2-7b-chat-hf | gpt-4-1106-preview | 0.340 | 0.178 | November 2023 |
| Llama-3.1-8B-Instruct | gpt-4-turbo | 0.645 | 0.528 | April 2024 |
| Llama-3.1-8B-Instruct | gpt-4o | 0.740 | 0.582 | July 2024 |
| — | gpt-4-turbo | 0.768 | 0.391 | April 2024 |

*Table 4.* Point in time results for MINION$\mathcal{S}$ configurations with best-in-class LocalLM and RemoteLM

| Total Chunks In-Context | Accuracy |
|:---:|:---:|
| 1 | 0.59375 |
| 16 | 0.53906 |
| 32 | 0.50000 |
| 64 | 0.48438 |
| 128 | 0.46094 |

*Table 5.* Accuracy vs. Number of Chunks in Context
Each chunk has 512 tokens.

| Number of Sub Tasks | Accuracy |
|:---:|:---:|
| 1 | 0.70313 |
| 2 | 0.39844 |
| 3 | 0.19531 |
| 4 | 0.14844 |

*Table 6.* Accuracy vs. Number of Sub Tasks

### E.3. Relationship with Retrieval-Augmented Generation

In this section, we discuss the relationship between local-remote collaboration and retrieval-augmented generation (RAG), a technique that reduces the number of tokens processed by an LM by retrieving a subset of relevant documents or chunks LM (Lewis et al., 2020; Karpukhin et al., 2020; Lee et al., 2019).

Retrieval-augmented generation and local-remote collaboration (*e.g.* MINION$\mathcal{S}$) are complementary techniques. They both provide a means to reduce cost by providing an LLM with a partial view of a large context. But, as we discuss below, they also have different error profiles and can be used in conjunction to improve performance.

#### E.3.1. COMPARISON OF MINION$\mathcal{S}$ AND RAG ON FINANCEBENCH

In Figure 8 (left), we plot the quality-cost trade-off on FINANCEBENCH for local-remote systems (MINION and MINION$\mathcal{S}$) and RAG systems using BM25 and OpenAI's `text-embedding-3-small` embeddings (Robertson & Zaragoza, 2009; Neelakantan et al., 2022). For RAG, we use a chunk size of 1000 characters, which we found to be optimal for this dataset after sweeping over chunk sizes with the BM25 retriever (see Figure 8 (center)). We show how a simple hyperparameter (number of retrieved chunks provided to the remote model) allows us to trade off quality of the RAG system for remote cost. Furthermore, we note that when the BM25 RAG system provides 50 or more chunks of the document to the remote model, it exceeds the performance of the remote model with the full context. This likely indicates that RAG helps in minimizing distractions from the long context. For FINANCEBENCH, when compared to MINION$\mathcal{S}$, the RAG system with OpenAI embeddings reaches similar points in the quality-cost trade-off space. Interestingly however, none of the RAG configurations are are able to match the quality of MINION at the same low cost.

#### E.3.2. COMPARISON OF MINION$\mathcal{S}$ AND RAG (EMBEDDINGS + BM25) ON SUMMARIZATION TASKS

In this section, we discuss the relationship between local-remote collaboration (e.g. MINION$\mathcal{S}$) and retrieval-augmented generation (RAG). These are complementary techniques with different strengths, which can be combined.

We perform a quantitative and qualitiateive compartive analysis demonstrating the different benefits of the two techniques. We analyze MINION$\mathcal{S}$, MINION, and RAG on two data intensive reasoning tasks: one that relies heavily on extraction (FINANCEBENCH) and another that requires long-document summarization (BOOOKSCORE). Our results demonstrate that, on FINANCEBENCH (see App. E.3.1), RAG can achieve similar, sometimes better, cost-accuracy tradeoffs compared to MINION$\mathcal{S}$, but is less cost-effective than MINION. On BOOOKSCORE (see App. E.3.2) — which requires integrating information dispersed across the document — we find that MINION$\mathcal{S}$ generates summaries comparable to a GPT-4O-only baseline when evaluated using an LLM-as-a-judge (see App. Table 10), while RAG results in lower quality summaries that miss critical plotlines, characters and details (see App. Table 11).

| Protocol | Local Model | Remote Model | FINANCEBENCH | | | | LONGHEALTH | | | | QASPER | | | |
|---|---|---|---|---|---|---|---|---|---|---|---|---|---|---|
| | | | Acc. | Cost | In Tok. (1k) | Out Tok. (1k) | Acc. | Cost | In Tok. (1k) | Out Tok. (1k) | Acc. | Cost | In Tok. (1k) | Out Tok. (1k) |
| Remote Only | — | GPT-4O | 0.826 | $0.261 | 103.04 | 0.32 | 0.748 | $0.301 | 120.10 | 0.07 | 0.598 | $0.137 | 54.40 | 0.09 |
| Local Only | LLAMA-8B | — | 0.326 | $0.000 | 0.00 | 0.00 | 0.468 | $0.000 | 122.58 | 0.07 | 0.538 | $0.000 | 54.41 | 0.06 |
| Local Only | LLAMA-1B | — | 0.000 | $0.000 | 0.00 | 0.00 | 0.115 | $0.000 | 122.58 | 0.07 | 0.000 | $0.000 | 54.41 | 0.10 |
| Local Only | LLAMA-3B | — | 0.130 | $0.000 | 0.00 | 0.00 | 0.345 | $0.000 | 122.58 | 0.08 | 0.164 | $0.000 | 54.41 | 0.08 |
| Local Only | QWEN-3B | — | 0.087 | $0.000 | 0.00 | 0.00 | 0.177 | $0.000 | 31.24 | 0.08 | 0.156 | $0.000 | 32.58 | 0.08 |
| MINION | LLAMA-8B | GPT-4O | 0.804 | $0.007 | 0.88 | 0.46 | 0.635 | $0.010 | 1.85 | 0.50 | 0.450 | $0.007 | 0.92 | 0.42 |
| MINION | LLAMA-3B | GPT-4O | 0.698 | $0.010 | 1.74 | 0.52 | 0.482 | $0.009 | 1.56 | 0.47 | 0.372 | $0.011 | 2.26 | 0.53 |
| MINION | QWEN-3B | GPT-4O | 0.217 | $0.029 | 8.28 | 0.82 | 0.281 | $0.021 | 5.70 | 0.68 | 0.210 | $0.035 | 10.51 | 0.87 |
| MINION$\mathcal{S}$ | LLAMA-8B | GPT-4O | 0.804 | $0.053 | 15.99 | 1.29 | 0.740 | $0.054 | 18.96 | 0.65 | 0.582 | $0.019 | 5.10 | 0.61 |
| MINION$\mathcal{S}$ | LLAMA-3B | GPT-4O | 0.726 | $0.079 | 24.67 | 1.77 | 0.703 | $0.057 | 20.11 | 0.66 | 0.558 | $0.020 | 5.62 | 0.60 |
| MINION$\mathcal{S}$ | QWEN-3B | GPT-4O | 0.783 | $0.059 | 17.20 | 1.56 | 0.645 | $0.043 | 14.43 | 0.65 | – | – | – | – |
| MINION$\mathcal{S}$ | QWEN-7B | GPT-4O | – | – | – | – | – | – | – | – | 0.600 | $0.015 | 3.44 | 0.61 |

*Table 7.* **Accuracy and cost of local-remote systems.** Evaluation of cost and accuracy on the FINANCEBENCH (Islam et al., 2023), LONGHEALTH (Adams et al., 2024), and QASPER (Dasigi et al., 2021). The table compares two edge-remote communication protocols — Naïve (Section 4) and MINION$\mathcal{S}$ (Section 5) — alongside edge-only and remote-only baselines. Three different edge models are considered (LLAMA-8B, LLAMA-3B, QWEN-3B, and LLAMA-1B) and a remote model (GPT-4O). Accuracy (Acc.) is the fraction of correct predictions across the dataset. Cost (USD) is the average cost in USD per query in the dataset computed. Costs are incurred for any calls to the remote model at GPT-4O rates (January 2025: $2.50 per million input tokens and $10.00 per million output tokens). We assume that running the edge model is free; see Section 3 for details on the cost model. In tokens is the number of input (*i.e.* prefill) tokens sent to the remote model. Out tokens is the number of output (*i.e.* decode) tokens generated from the remote model. Both values are shown in thousands.

| Protocol | Local Model | Remote Model | Macro Avg. | | Information Extraction | | Numerical Reasoning | |
|---|---|---|---|---|---|---|---|---|
| | | | Acc. | Cost | Acc. | Cost | Acc. | Cost |
| Remote Only | — | GPT-4O | 0.883 | $0.259 | 1.000 | $0.246 | 0.767 | $0.271 |
| Local Only | LLAMA-8B | — | 0.367 | $0.000 | 0.467 | $0.000 | 0.267 | $0.000 |
| Local Only | LLAMA-1B | — | 0.000 | $0.000 | 0.000 | $0.000 | 0.000 | $0.000 |
| Local Only | LLAMA-3B | — | 0.150 | $0.000 | 0.200 | $0.000 | 0.100 | $0.000 |
| Local Only | QWEN-3B | — | 0.083 | $0.000 | 0.067 | $0.000 | 0.100 | $0.000 |
| MINION | LLAMA-8B | GPT-4O | 0.850 | $0.007 | 0.933 | $0.006 | 0.767 | $0.007 |
| MINION | LLAMA-3B | GPT-4O | 0.743 | $0.010 | 0.831 | $0.013 | 0.656 | $0.007 |
| MINION | QWEN-3B | GPT-4O | 0.233 | $0.028 | 0.267 | $0.025 | 0.200 | $0.032 |
| MINION$\mathcal{S}$ | LLAMA-8B | GPT-4O | 0.840 | $0.048 | 0.893 | $0.031 | 0.787 | $0.064 |
| MINION$\mathcal{S}$ | LLAMA-3B | GPT-4O | 0.730 | $0.077 | 0.693 | $0.067 | 0.767 | $0.087 |
| MINION$\mathcal{S}$ | QWEN-3B | GPT-4O | 0.793 | $0.055 | 0.773 | $0.044 | 0.813 | $0.067 |

*Table 8.* **Stratified analysis of local-remote systems on FINANCEBENCH.** Evaluation of cost and accuracy on the FINANCEBENCH (Islam et al., 2023) stratified by question type. We consider two types of questions: *Information Extraction*, questions that require extracting specific data or textual content from filings, and, Numerical Reasoning, which requires performing additional numerical reasoning and calculations on extracted data. The table compares two edge-remote communication protocols — MINION (Section 4) and MINION$\mathcal{S}$ (Section 5) — alongside edge-only and remote-only baselines. Three different edge models are considered (LLAMA-8B, LLAMA-3B, QWEN-3B, and LLAMA-1B) and a remote model (GPT-4O). Accuracy (Acc.) is the fraction of correct predictions across the dataset. Cost (USD) is the average cost in USD per query in the dataset computed. Costs are incurred for any calls to the remote model at GPT-4O rates (January 2025: $2.50 per million input tokens and $10.00 per million output tokens). We assume that running the edge model is free; see Section 3 for details on the cost model.

| Protocol | Local Model | Remote Model | Macro Avg. | | 1 evidence span | | 2 evidence spans | | 3 evidence spans | | 4+ evidence spans | |
|---|---|---|---|---|---|---|---|---|---|---|---|---|
| | | | Acc. | Cost | Acc. | Cost | Acc. | Cost | Acc. | Cost | Acc. | Cost |
| Remote Only | — | GPT-4O | 0.668 | $0.301 | 0.815 | $0.301 | 0.661 | $0.302 | 0.600 | $0.302 | 0.596 | $0.301 |
| Local Only | LLAMA-8B | — | 0.448 | $0.000 | 0.498 | $0.000 | 0.390 | $0.000 | 0.520 | $0.000 | 0.386 | $0.000 |
| Local Only | LLAMA-1B | — | 0.097 | $0.000 | 0.124 | $0.000 | 0.102 | $0.000 | 0.040 | $0.000 | 0.123 | $0.000 |
| Local Only | LLAMA-3B | — | 0.327 | $0.000 | 0.375 | $0.000 | 0.271 | $0.000 | 0.400 | $0.000 | 0.263 | $0.000 |
| Local Only | QWEN-3B | — | 0.174 | $0.000 | 0.185 | $0.000 | 0.186 | $0.000 | 0.200 | $0.000 | 0.123 | $0.000 |
| MINION | LLAMA-8B | GPT-4O | 0.582 | $0.011 | 0.683 | $0.009 | 0.627 | $0.009 | 0.560 | $0.017 | 0.456 | $0.011 |
| MINION | LLAMA-3B | GPT-4O | 0.436 | $0.009 | 0.529 | $0.008 | 0.390 | $0.009 | 0.440 | $0.008 | 0.386 | $0.010 |
| MINION | QWEN-3B | GPT-4O | 0.259 | $0.019 | 0.298 | $0.023 | 0.288 | $0.016 | 0.240 | $0.017 | 0.211 | $0.018 |
| MINION$\mathcal{S}$ | LLAMA-8B | GPT-4O | 0.731 | $0.056 | 0.753 | $0.053 | 0.763 | $0.051 | 0.760 | $0.060 | 0.649 | $0.060 |
| MINION$\mathcal{S}$ | LLAMA-3B | GPT-4O | 0.665 | $0.058 | 0.741 | $0.056 | 0.661 | $0.055 | 0.680 | $0.061 | 0.579 | $0.062 |
| MINION$\mathcal{S}$ | QWEN-3B | GPT-4O | 0.635 | $0.045 | 0.649 | $0.041 | 0.678 | $0.040 | 0.600 | $0.050 | 0.614 | $0.048 |

*Table 9.* **Stratified analysis of local-remote systems on LONGHEALTH.** Evaluation of cost and accuracy on LONGHEALTH (Adams et al., 2024) stratified by the *dispersion* of the supporting evidence for each question. We measure dispersion with the number of distinct spans in the context which contain information relevant to the question. We include performance for questions with 1, 2, 3, and 4+ distinct evidence spans. The table compares two edge-remote communication protocols — MINION (Section 4) and MINION$\mathcal{S}$ (Section 5) — alongside edge-only and remote-only baselines. Three different edge models are considered (LLAMA-8B, LLAMA-3B, QWEN-3B, and LLAMA-1B) and a remote model (GPT-4O). Accuracy (Acc.) is the fraction of correct predictions across the dataset. Cost (USD) is the average cost in USD per query in the dataset computed. Costs are incurred for any calls to the remote model at GPT-4O rates (January 2025: $2.50 per million input tokens and $10.00 per million output tokens). We assume that running the edge model is free; see Section 3 for details on the cost model.

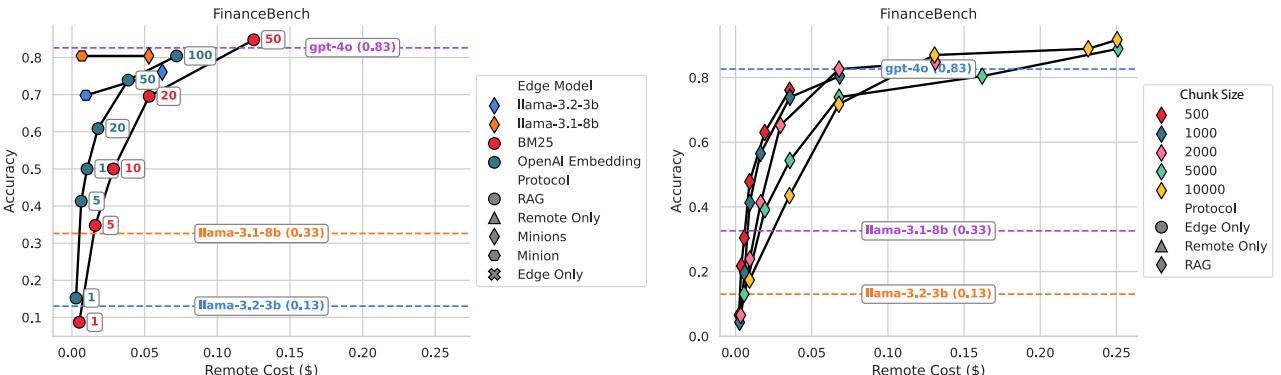

*Figure 8.* **Relationship with retrieval-augmented generation.**

RAG is a very suitable approach for FINANCEBENCH, since all of the questions heavily rely on information extraction from specific sections of financial statements. However, RAG will not be suitable for a summarization task, unlike small LMs. Therefore, we use the long-document summarization dataset, BOOOOKSCORE (Chang et al., 2023). BOOOOKSCORE which contains a set of 400 books published between 2023-2024. The average story length in BOOOOKSCORE is 128179 tokens with a max of 401486 tokens and a minimum of 26926 tokens. We utilize both MINION$\mathcal{S}$, RAG (w/Embeddings + BM25), and GPT-4O only to complete the task. We describe the set-up for all three approaches next.

**MINION$\mathcal{S}$ for summarization** In applying MINION$\mathcal{S}$ to the task, the LocalLM (LLAMA-3.2-3B-Instruct) provides summaries on chunks of the original text, passing a list of chunk summaries to the RemoteLM (GPT-4O). RemoteLM produces the final summary.

**RAG (Embedding) for summarization** In our embedding-based RAG approach, we use the OpenAI TEXT-EMBEDDING-3-SMALL to embed chunks of the original text (of length 5000 characters) and we retrieve the top-15 most relevant chunks using the query "Summarize the provided text". We then prompt GPT-4O to generate a complete summary over the retrieved chunks.

**RAG (BM25) for summarization** In our BM25-based RAG approach, we use the BM25 to retrieve chunks of the original

text (of length 5000 characters) based on the query: "Summarize the provided text". We retrieve the top-15 most relevant chunks and prompt GPT-4O to produce a final summary over the retrieved chunks. We choose top-15 to ensure the number of tokens passed up by the baseline is comparable with those passed up by MINION$\mathcal{S}$.

**GPT-4o** In our final baseline, we use GPT-4O alone to create the story summaries. For texts that extend beyond the 128K context length window, we truncate the stories.

### Evaluation

- **Qualitative** In Table 11 we provide samples outputs from each of the 4 methods described above. We highlight major events in red, themes in green, locations in blue and names in indigo. The samples demonstrate that amongst all the methods, MINION$\mathcal{S}$ outputs contain the most entity mentions and story specific details. Moreover, when compared to GPT-4O-only RemoteLM, MINION$\mathcal{S}$ is $9.3\times$ more efficient — 11,500 versus the full 108,185 prefill tokens.

  The summaries from MINION$\mathcal{S}$ are generally $1.3\times$ longer and more verbose than the RAG systems' summaries, likely indicating that the former is more effective at "passing forward" salient information. Moreover, RAG systems' summaries are missing the main arc of the narrative in favor of what seems an assortment of facts.

- **Quantitative** We additionally perform a quantitative analysis of the generated summaries using a LLM-as-a-judge framework. As an evaluator, we use the CLAUDE-3.5-SONNET model, to avoid any biases between the evaluator and the supervisor model. We prompt the model with the generated summary, ground truth summary (gpt4-4096-inc-cleaned) provided from the original BOOOOKSCORE generations, and a grading rubric (see Figure 9). The rubric evaluates 7 criteria: coherence, relevance, conciseness, comprehensiveness, engagement & readability, accuracy, and thematic depth. We prompt CLAUDE-3.5-SONNET to generate a score (1-5) for each of the criteria and average the scores. We find that summaries generated by MINION$\mathcal{S}$ score comparably with GPT4O-only generated summaries, while RAG based baselines perform worse. Our results can be found in Table 10.

---

**Evaluation Rubric for Summaries**

1. **Coherence (1-5):** Summary is logically structured, with clear connections between events, avoiding abrupt jumps or inconsistencies.

2. **Relevance (1-5):** Accurately reflects key themes, events, and characters, focusing on essential details without unnecessary plot points.

3. **Conciseness (1-5):** Thorough yet avoids excessive detail, presenting necessary information without redundancy.

4. **Comprehensiveness (1-5):** Covers all major characters, events, and themes, ensuring a complete overview without omissions.

5. **Engagement & Readability (1-5):** Engaging and easy to read, with well-constructed sentences and clear, precise language.

6. **Accuracy (1-5):** Stays true to the book's storyline, themes, and tone, with correct details, names, and events.

7. **Thematic Depth (1-5):** Identifies underlying themes and messages, providing insights into conflicts, motivations, and resolutions.

---

*Figure 9.* Evaluation Rubric for Summaries

| Method | Score |
|---|---|
| MINION$\mathcal{S}$ | 3.01 |
| GPT4o | 3.06 |
| RAG (BM25) | 2.48 |
| RAG (Embedding) | 2.38 |

*Table 10.* Comparison of Methods and Rubric Scores

Table 11: Comparative Analysis of Story Summaries Across Different Methods

| Story Title | Minions | GPT-4o | BM25 | Embedding |
|---|---|---|---|---|
| *A Likely Story* | The novel follows the complex dynamics of the Manning family, focusing on Isabelle Manning, a struggling writer grappling with the legacy of her famous father, Ward Manning, and the recent death of her mother, Claire. As Isabelle navigates her career and personal life, she uncovers family secrets, including her father's plagiarism and her mother's hidden manuscript, which challenge her identity and ambitions. The narrative explores themes of fame, family expectations, and personal growth, culminating in Isabelle's journey towards self-discovery and independence. Set against the backdrop of New York City and Sag Harbor, the story delves into the intricacies of familial relationships and the pursuit of creative fulfillment. | Isabelle Manning, daughter of a famous author, discovers an unfinished manuscript by her late mother, Claire. She completes and publishes it as her own, leading to success but also guilt and exposure. Her father, Ward, reveals he sabotaged her first book to teach her resilience, deepening her sense of betrayal. Ultimately, Isabelle confesses her actions, seeks redemption, and begins to redefine her identity beyond her father's shadow. | Isabelle Manning, overshadowed by her famous father Ward, receives a threatening text, "I KNOW WHAT YOU DID," hinting at a secret tied to her book, "Underpainting." Meanwhile, her mother Claire's past support and her father's hidden illness add layers to Isabelle's challenges. | In a narrative spanning New York and Sag Harbor, Claire Cunningham grapples with her identity beyond being a Vassar graduate, while her daughter Isabelle faces personal and professional challenges, including her father's illness and her own writing struggles. |
| *All the Dangerous Things* | Isabelle Drake, a woman grappling with the traumatic disappearance of her son Mason, navigates a complex web of grief, guilt, and suspicion. As she becomes entangled with true crime enthusiasts and investigators, including podcast host Waylon and Detective Dozier, Isabelle's quest for truth reveals unsettling family secrets and personal betrayals. Her journey is marked by strained relationships, particularly with her ex-husband Ben and his connections to other women, including Valerie and Allison. Throughout the narrative, themes of motherhood, mental health, and societal judgment are explored, culminating in a deeper understanding of the pressures and expectations faced by women. | Isabelle Drake, plagued by insomnia and guilt, is desperate to find her missing son, Mason. She suspects her husband, Ben, and his new partner, Valerie. With Waylon's help, she discovers Abigail Fisher, manipulated by Valerie, took Mason believing she was rescuing him. | The narrative follows Isabelle, dealing with Mason's disappearance. She works with podcaster Waylon, uncovering links to Ben's deceased wife, Allison. | Isabelle, struggling with grief and insomnia, joins a grief counseling group. She meets Valerie and collaborates with Waylon, but becomes wary after finding unsettling information on his laptop. |
| | | | | *Continued on next page* |

| | | *Continued from previous page* | | |
|---|---|---|---|---|
| **Story Title** | **Minions** | **GPT-4o** | **BM25** | **Embedding** |
| *A Living Remedy: A Memoir* | Nicole Chung, a Korean American adoptee, reflects on her complex relationships with her adoptive parents, her identity, and the challenges of navigating life as a minority in a predominantly white community in Oregon. Her memoir explores themes of family, loss, and resilience, as she recounts her father's death from kidney failure, and her mother's battle with cancer. Amidst these personal challenges, Chung grapples with her own grief, financial struggles, and the impact of the COVID-19 pandemic, while finding solace in her family, faith, and writing. Her journey is marked by a deep appreciation for her parents' sacrifices, the support of her husband and children, and the enduring legacy of love and forgiveness instilled by her mother. | Nicole Chung's memoir explores her journey after the loss of her adoptive parents. As a Korean adoptee, she reflects on family's financial struggles, parents' health battles, and their deaths' impact on her identity. She finds solace in writing and her own family. | The protagonist struggles with visiting her dying mother during the COVID-19 pandemic. The story explores grief, family responsibility, and cherishing life amidst adversity. | A woman reflects on her parents' illnesses and deaths, balancing her role as a daughter and mother. She finds solace in childhood memories and the legacy of her parents' love. |
| *A House with Good Bones* | Samantha, a 32-year-old archaeoentomologist, returns to her childhood home on Lammergeier Lane in North Carolina, where she confronts her family's dark past, including her grandmother Gran Mae's mysterious and malevolent legacy. As Samantha navigates her mother's strange behavior and the eerie presence of vultures, she uncovers secrets involving ritual magic, a jar of human teeth, and the supernatural "underground children." With the help of her friend Gail and handyman Phil, Samantha faces the haunting manifestations of her family's history. The novel explores themes of family, memory, and the supernatural, blending elements of horror and fantasy. | Samantha Montgomery returns home to find her mother acting strangely and the house devoid of insects. She uncovers a dark history involving her great-grandfather, a sorcerer, and her grandmother, who used roses to wield power. With help from Gail and Phil, she confronts the terrifying "underground children," using rose power to banish threats. | The protagonist returns to their grandmother's unchanged garden, filled with roses but mysteriously devoid of insects. They uncover unsettling truths about their grandmother's past and their mother's current state of mind. The narrative explores themes of family legacy and the passage of time. | Samantha, an archaeoentomologist, returns to her childhood home and finds herself investigating insect collections. Dealing with sleep paralysis and memories of her grandmother, she discovers the peculiar absence of insects in the garden. She navigates family dynamics and her mother's anxiety amid an eerie atmosphere. |

## E.4. Energy Consumption Analysis

We perform an energy analysis of MINION$\mathcal{S}$ workloads by measuring the total energy consumed by LocalLM and *estimating* the energy consumed by RemoteLM, using Joules (J) as the primary unit of measurement. For our RemoteLM energy consumption estimates, we use the inference-time energy estimate calculations provided by Epoch AI (AI, 2024). We perform our analysis running our local workloads on two types of hardware: an A100 Nvidia GPU and Apple M1 Max GPU. For measuring energy consumption on these hardware set-ups, we use off the shelf packages such as powermetrics and nvidia-smi. We sweep our experiments across two LocalLM variants (LLAMA-3.2-1B and LLAMA-3.2-3B) and use GPT-4O as the RemoteLM. As a baseline, we estimate the cost of entirely running the MINION$\mathcal{S}$ workload on RemoteLM estimating the total energy consumption that GPT-4O would have if it performed the equivalent work (as measured by input / output tokens) of the LocalLM. We compare energy consumption between MINION$\mathcal{S}$ and remote-only setups, varying the length of the input context length. Our results indicate that we can achieve up to $12\times$ energy consumption savings when using a $1B$ parameter local model and up to $6\times$ savings with a $3B$ parameter model running local workloads on an A100. Similarly, we see energy consumption reductions when running workloads on the M-series hardware of up to $8\times$. Please see Figure 10 for more details.

## E.5. Local LM Agentic Tool Use

We take a preliminary step in extending MINION$\mathcal{S}$ to support LocalLM tool calling. This configuration enables the LocalLM to autonomously execute tools necessary for task completion, guided by the RemoteLM awareness of available local tools (i.e., the RemoteLM might request the LocalLM to describe the contents of the local filesystem, further suggesting it use the tool that lists all local files). We evaluate this extended setup on a hand-curated set of queries (see Table 13 for sample queries) against a local folder containing reciepts, giving the LocalLM access to the following tools: listDirectory, readFile, readMultipleFiles, writeFile, createDirectory, and searchFiles. For the purposes of our evaluation, we use QWEN2.5-3B and QWEN2.5-1B as our LocalLM's and GPT-4O as the RemoteLM. We compare performance against a remote-only baseline which answers the queries given *all* the documents in context. We find that when using QWEN2.5-3B locally, MINION$\mathcal{S}$ performance is comparable to the GPT-4O baseline, will only using $< 28\%$ of remote token usage (see Listing 1 for a sample Minion trace with tool calling enabled). However, we find that QWEN2.5-1B is unable to achieve parity in performance and lacks the ability to effectively utilize tools. See Table 12 for complete results.

```
1 {
2   "task": "Do I have a receipt for Guapas? If yes, please tell me the total amount.",
```

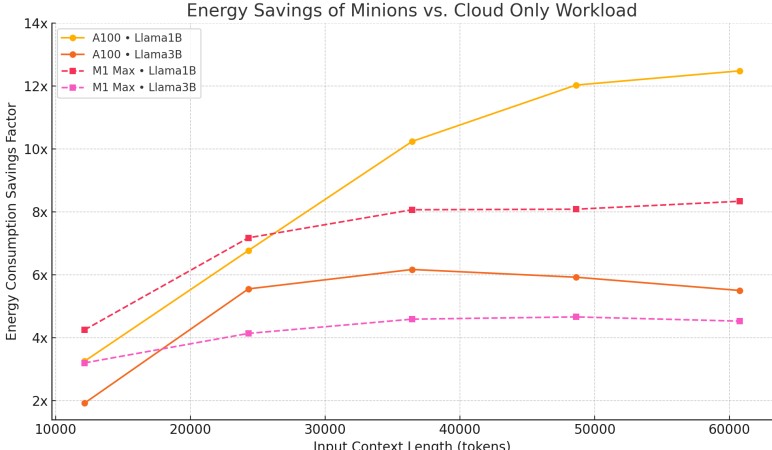

*Figure 10.* **Energy savings of Minions vs. cloud-only workload across varying input context lengths** We compare the energy consumption of local execution of Minion using two hardware configurations—NVIDIA A100 and Apple M1 Max—against a cloud-only baseline powered by GPT-4O. Local workloads are run using LLAMA-3.2-1B and LLAMA-3.2-3B models, with energy measured via nvidia-smi and powermetrics, while remote energy use is estimated following methodology from Epoch AI (AI, 2024). Across both hardware types, local execution yields substantial energy savings, reaching up to 12× with the 1B model on A100, and consistent gains across other configurations.

| Local Model | Remote Model | Task Completion Rate | Total Remote Tokens Processed |
|---|---|---|---|
| QWEN2.5-1.5B | GPT-4O | 0.10 | 29697.6 |
| QWEN2.5-3B | GPT-4O | **0.70** | **8741.1** |
| — | GPT-4O | **0.70** | 32498.3 |

*Table 12.* Comparison of task completion rates and token costs: the MINION𝒮 set-up performs comparable with the GPT-4O-only baseline while reducing RemoteLM prefill costs by 3.7×.

| Question | Answer |
|---|---|
| Tell me all the dates I ordered sprouts | 3/5, 1/26, 1/29, 2/10, 9/24 |
| I am tracking my calories. Did I order a tiramisu at Terun? | Yes |
| How many times did I eat at Sultana? | 3 |
| What is the most common menu item ordered across all my Asian Box receipts? | The Workout |

*Table 13.* Queries for agentic tool use analysis: sample question/answer pairs for the reciept history question-answering task used in Section E.5.

```
 3    "steps" : [
 4        {
 5          "user": "remote",
 6          "prompt": "We need to perform the following task.\n\n### Task\nDo I have a receipt for Guapas? If yes, please tell me the
            total amount.\n\n### Instructions\nYou will not have direct access to the context, but can chat with a small language model
            has access to local context via the following tools:\n...\n",
 7          "output": "{\"message\":\Please search for files related to guapas: "search_files(path='.', pattern='*Guapas*',
          exclude_patterns=['*.jpg', '*.png', '*.gif'])\"}"
 8        },
 9        {
10          "user": "local",
11          "prompt": "search_files(path='.', pattern='*Guapas*', exclude_patterns=['*.jpg', '*.png', '*.gif'])",
12          "output": ""
13        },
14        {
15          "user": "tool",
16          "tool": "search_files",
17          "arguments": {
```

```
18          "exclude_patterns": ["*.jpg", "*.png", "*.gif"],
19          "path": ".",
20          "pattern": "*Guapas*"
21        },
22        "output": "Found 'guapas.pdf' in the current directory. Not necessarily a receipt."
23      },
24      {
25        "user": "remote",
26        "prompt": "Analyze the response from the local language model...",
27        "output": "1. We found 'guapas.pdf'.\n2. Not sufficient. Need to read the file.\n3. Missing content.\n4. Next tool:
      read_file.\n\n{\"message\": \"read_file(file_path='guapas.pdf')\"}"
28      },
29      {
30        "user": "remote",
31        "prompt": "Response after step-by-step thinking...",
32        "output": "{\"message\": \"read_file(file_path='guapas.pdf')\"}"
33      },
34      {
35        "user": "remote",
36        "prompt": "If task is complete, provide final answer. Otherwise, request additional info.",
37        "output": "{ \"decision\": \"request_additional_info\", \"message\": \"Please read the file 'guapas.pdf' to determine if
      it is a receipt and, if so, extract the total amount mentioned in it.\" }"
38      },
39      {
40        "user": "local",
41        "prompt": "Please read the file 'guapas.pdf' to determine if it is a receipt and, if so, extract the total amount
      mentioned in it.",
42        "output": ""
43      },
44      {
45        "user": "tool",
46        "tool": "read_file",
47        "arguments": {
48          "file_path": "guapas.pdf"
49        },
50        "output": "The file contains a DoorDash receipt for Guapas Mexican Grill. Total Amount: $659.57"
51      },
52      {
53        "user": "remote",
54        "prompt": "Analyze the final tool output and determine if sufficient.",
55        "output": "1. 'guapas.pdf' is a receipt.\n2. Information is sufficient.\n3. No missing info.\n4. Final Answer:\n\n{\"
      decision\": \"provide_final_answer\", \"answer\": \"Yes, you have a receipt for Guapas. The total amount is $659.57.\"}"
56      },
57      {
58        "user": "remote",
59        "prompt": "Final confirmation step",
60        "output": "{\"decision\": \"provide_final_answer\", \"answer\": \"Yes, you have a receipt for Guapas. The total amount is
       $659.57.\"}"
61      }
62    ]
63  }
```

*Listing 1.* Full Minion trace with local LLM tool calling

## E.6. Analysis of Network latency

We measure network latency in the MINION setup. In our evaluation, the MINION configuration uses LocalLM as LLAMA-3.2-1B and RemoteLM as GPT-4O. For each round of communication, we record the time spent on calls to LocalLM and RemoteLM, as well as the time spent on communication between the local and remote clients. Across 15 queries, each with an input context length of 12,234 tokens, we find that communication time accounts for less than $< 0.002\%$ of the total time per round.

## E.7. Extension to Multimodal Tasks

We demonstrate the extension of MINIONS to multimodal settings. Specifically, we evaluate on a multimodal math question-answering task: MATH-VISION (Wang et al., 2024b). MATH-VISION is a challenging dataset that tests a wide range of mathematical skills, ranging from algebra to geometry to combinatorics, which require reasoning over visual input (i.e., diagrams). For our setup, we set LocalLM to be GEMMA3-4B (a SOTA VLM) and use GPT-4O as the RemoteLM. For our MINIONS implementation, the image stays local and is not seen by RemoteLM. For our remote-only baseline, the image and question are passed directly to GPT-4O. On a sample of the test set, we find that MINIONS is within 10 points of the GPT-4O-only baseline – 18.2% vs 27.8% accuracy.

# F. Prompts

## F.1. MINION

RemoteLM

```
1  We need to answer the following question based on a {doc_type}.
2
3  ### Question
4  {query}
5
6  ### Instructions
7  You will not have direct access to the {doc_type}, but can chat with a small language model which has
        read the entire thing.
8
9  Feel free to think step-by-step, but eventually you must provide an output
10 in the format below:
11
12 <think step by step here>
13 ```json
14 {{
15     "message": "<your message to the small language model>"
16 }}
17 ```
```

LocalLM

```
1  You will help a user answer the following question based on a {doc_type}.
2
3
4  Read the {doc_type} below and prepare to answer questions from an expert user.
5  ### {doc_type}
6  {context}
7
8  ### Question
9  {query}
```

Conversation

```
1  Here is the response from the small language model:
2
3  ### Response
4  {response}
5
6
7  ### Instructions
8  Analyze the response and think-step-by-step to determine if you have enough
9  information to answer the question.
10
11 If you have enough information, provide a final numeric answer in the format
12 below.
13
14 <think step by step here>
15 ```json
16 {{
17     "decision": "provide_final_answer",
18     "answer": "<your answer>"
19 }}
20 ```
21
22 Otherwise, request additional information from the small language model by
23 outputting the following:
24
25 <think step by step here>
26 ```json
27 {{
28     "decision": "request_additional_info",
29     "message": "<your message to the small language model>"
30 }}
31 ```
```

## F.2. MINION𝒮

### MINION𝒮: FINANCEBENCH

Decompose

```
 1 # Decomposition Round #{step_number}
 2
 3 You do not have access to the raw document(s), but instead can assign tasks to small and less capable
        language models that can read the document(s).
 4 Note that the document(s) can be very long, so each task should be performed only over a small chunk of
        text.
 5
 6 Write a Python function that will output formatted tasks for a small language model.
 7 Make sure that NONE of the tasks require calculations or complicated reasoning.
 8 Any information you mentioned in your task should be given an extraction task.
 9
10 Please use chunks of {pages_per_chunk} pages using the `chunk_on_multiple_pages(doc = context,
        pages_per_chunk ={pages_per_chunk})` function.
11
12 If you have multiple tasks, consider using nested for-loops to apply a set of tasks to a set of chunks.
         Though it's not required to have more than one task.
13
14 {ADVANCED_STEPS_INSTRUCTIONS}
15
16 Assume a Pydantic model called `JobManifest(BaseModel)` is already in global scope. For your reference,
         here is the model:
17 ```
18 {manifest_source}
19 ```
20 Assume a Pydantic model called `JobOutput(BaseModel)` is already in global scope. For your reference,
         here is the model:
21 ```
22 {output_source}
23 ```
24 DO NOT rewrite or import the model in your code.
25
26 The function signature will look like:
27 ```
28 {signature_source}
29 ```
30
31 You can assume you have access to the following chunking function(s). Do not reimplement the function,
         just use it.
32 ```
33 {chunking_source}
34 ```
```

Worker

```
 1 Your job is to complete the following task using only the context below. The context is a chunk of text
        taken arbitrarily from a document, it might or might not contain relevant information to the task.
 2
 3 ## Document
 4 {context}
 5
 6 ## Task
 7 {task}
 8
 9 {advice}
10
11 Return your result in JSON with the following keys: "explanation", "citation", and "answer".
12
13 - "explanation": A concise statement of your reasoning or how you concluded your answer.
14 - "citation": A direct snippet of the text that supports your answer. If nothing is found, put "None".
15 - "answer": The extracted answer. If nothing is found, put "None".
16
17 Be certain to only rely on the provided text. If you cannot determine the information confidently from
        this chunk, respond with "None" for all fields.
```

Synthesize

```
1
2  Now synthesize the findings from multiple junior workers (LLMs).
3  Your task is to finalize an answer to the question below **if and only if** you have sufficient,
       reliable information.
4  Otherwise, you must request additional work.
5
6  ---
7  ## Inputs
8  1. Question to answer:
9  {question}
10
11 2. Collected Job Outputs (from junior models):
12 {extractions}
13
14 ---
15 First think step-by-step and then answer the question using the exact format below.
16
17 ## ANSWER GUIDELINES
18 1. **Determine if the collected Job Outputs provide enough trustworthy, consistent evidence to
       confidently answer the question.**
19    - If the data is incomplete or contradictory, do NOT guess. Instead, specify what is missing.
20    - If the evidence is sufficient, provide a final answer.
21
22 2. **Be conservative.** When in doubt, ask for more information.
23
24 3. **Address conflicts.** If multiple jobs give different answers, rely on whichever is best supported
       by a valid "explanation" and "citation".
25    - If you need more information from the conflicting jobs you could request additional work from
      those specific jobs (be sure to mention the specific job IDs in your additional_info field).
26    - Then, in the next round you can make a smaller set of jobs to determine which answer is correct.
27
28 4. **Required JSON Output**: You must output a JSON object with these keys:
29    - "decision": Must be either "provide_final_answer" OR "request_additional_info".
30      - Use "provide_final_answer" if you have enough information.
31      - Use "request_additional_info" if you cannot conclusively answer.
32    - "explanation": A short statement about how you arrived at your conclusion or what is still missing
      .
33    - "answer": The final answer string if "decision"="provide_final_answer", or null otherwise. Should
      contain ONLY the final answer, without additional calculations or explanations.
34
35 Here is the template for your JSON response (with no extra text outside the JSON):
36
37 <think step-by-step here>
38 ```json
39 {{
40 "decision": "...",
41 "explanation": "...",
42 "answer": "... or null", # Good answer format: "0.56"; Bad answer format: "The ratio is calculated as
       1-0.27*2 = 0.56"
43 }}
44 ```
45
46 **Important**:
47 - If there is not enough information, set "answer" to null, set "decision" to "request_additional_info
      ", and specify exactly what else you need in "missing_info" and from which job IDs.
48
49 Now, carefully inspect the question, think step-by-step and perform any calculations before outputting
       the JSON object.
```

## MINION𝒮: LONGHEALTH

Decompose

```
1  # Decomposition Round #{step_number}
2
3  You do not have access to the raw document(s), but instead can assign tasks to small and less capable
       language models that can read the document(s).
4  Note that the document(s) can be very long, so each task should be performed only over a small chunk of
       text.
5
6  Write a Python function that will output formatted tasks for a small language model.
7  Make sure that NONE of the tasks require multiple steps. Each task should be atomic!
8  Consider using nested for-loops to apply a set of tasks to a set of chunks.
9  The same `task_id` should be applied to multiple chunks. DO NOT instantiate a new `task_id` for each
       combination of task and chunk.
10 Use the conversational history to inform what chunking strategy has already been applied.
11
12 {ADVANCED_STEPS_INSTRUCTIONS}
13
14 Assume a Pydantic model called `JobManifest(BaseModel)` is already in global scope. For your reference,
       here is the model:
15 ```
16 {manifest_source}
17 ```
18 Assume a Pydantic model called `JobOutput(BaseModel)` is already in global scope. For your reference,
       here is the model:
19 ```
20 {output_source}
21 ```
22 DO NOT rewrite or import the model in your code.
23
24 The function signature will look like:
25 ```
26 {signature_source}
27 ```
28
29
30 You can assume you have access to the following chunking function(S). Do not reimplement the function,
       just use it.
31 ```
32 {chunking_source}
33 ```
34
35 Here is an example
36 ```
37 task_id = 1  # Unique identifier for the task
38 for doc_id, document in enumerate(context):
39     # if you need to chunk the document into sections
40     chunks = chunk_by_section(document)
41     # or if you need to chunk the document into pages
42     chunks = chunk_by_page(document)
43
44     for chunk_id, chunk in enumerate(chunks):
45         # Create a task for extracting mentions of specific keywords
46         task = (
47             "Extract all mentions of the following keywords: "
48             "'Ca19-9', 'tumor marker', 'September 2021', 'U/ml', 'Mrs. Anderson'."
49         )
50         job_manifest = JobManifest(
51             chunk_id=f"doc_id_chunk_id",
52             task_id=task_id,
53             chunk=chunk,
54             task=task,
55             advice="Focus on extracting the specific keywords related to Mrs. Anderson's tumor marker
       levels."
56         )
57         job_manifests.append(job_manifest)
58 ```
```

**P**worker

```
 1 Your job is to complete the following task using only the context below. The context is a chunk of text
       taken arbitrarily from a document, it might or might not contain relevant information to the task.
 2
 3 ## Document
 4 {context}
 5
 6 ### Question you are trying to answer:
 7 {question}
 8
 9 # You have been instructed to extract information pertaining to the following concepts:
10 # \"Date of visit\", {task}
11
12 Format your response as follows:
13 {{
14 "Date of visit" : "`direct quote extracted text`",
15 "<keyword_1>" : "`direct quote extracted text`",
16 "<keyword_2>" : "`direct quote extracted text`",
17 ...
18 }}
19
20 Can you please extract the relevant sections from the document that are related to the concepts
       provided? Extract direct quotes or sentences. If concept is not mentioned, leave it out.
21
22 Your Answer:
```

**Psynthesize**

```
 1 Answer the following by the synthesizing findings from multiple junior workers (LLMs).
 2
 3
 4 ---
 5 ## Inputs
 6 1. Question to answer:
 7 {question}
 8
 9 2. Collected Job Outputs (from junior models):
10 {extractions}
11
12 ---
13 First think step-by-step and then answer the question using the exact format below.
14
15 ## ANSWER GUIDELINES
16
17 **Required JSON Output**: You must output exactly one JSON object with these keys:
18    - "decision": Must be  "provide_final_answer".
19    - "explanation": A short statement about how you arrived at your conclusion or what is still missing
       .
20    - "answer": The final answer string (that matches one of the provided options) if "decision"="
       provide_final_answer", or null otherwise.
21
22
23 Here is the template for your JSON response:
24
25 <think step-by-step here>
26
27
28 {{
29 "decision": "...",
30 "explanation": "...",
31 "answer": "...",
32 }}
33
34
35 Now, carefully inspect the question, think step-by-step and perform any calculations before outputting
       the JSON object. If answer choices are provided, your answer must **exactly** match one of the
       answer choices.
36
37 Question:
38 {question}
39
40 Your Answer:
```

# MINION$S$: QASPER

**P_decompose**

```
 1 # Decomposition Round #{step_number}
 2
 3 You do not have access to the raw document(s), but instead can assign tasks to small and less capable
       language models that can read the document(s).
 4 Note that the document(s) can be very long, so each task should be performed only over a small chunk of
        text.
 5
 6 Write a Python function that will output formatted tasks for a small language model.
 7 Make sure that NONE of the tasks require multiple steps. Each task should be atomic!
 8 Consider using nested for-loops to apply a set of tasks to a set of chunks.
 9 The same `task_id` should be applied to multiple chunks. DO NOT instantiate a new `task_id` for each
       combination of task and chunk.
10 Use the conversational history to inform what chunking strategy has already been applied.
11
12 {ADVANCED_STEPS_INSTRUCTIONS}
13
14 Assume a Pydantic model called `JobManifest(BaseModel)` is already in global scope. For your reference,
        here is the model:
15 ```
16 {manifest_source}
17 ```
18 Assume a Pydantic model called `JobOutput(BaseModel)` is already in global scope. For your reference,
       here is the model:
19 ```
20 {output_source}
21 ```
22 DO NOT rewrite or import the model in your code.
23
24 The function signature will look like:
25 ```
26 {signature_source}
27 ```
28
29
30 You can assume you have access to the following chunking function(S). Do not reimplement the function,
       just use it.
31 ```
32 {chunking_source}
33 ```
34
35 Here is an example
36 ```
37 task_id = 1  # Unique identifier for the task
38 for doc_id, document in enumerate(context):
39     # if you need to chunk the document into sections
40     chunks = chunk_by_section(document)
41     # or if you need to chunk the document into pages
42     chunks = chunk_by_page(document)
43
44     for chunk_id, chunk in enumerate(chunks):
45         # Create a task for extracting mentions of specific keywords
46         task = (
47             "Extract all mentions of the following keywords: "
48             "'Ca19-9', 'tumor marker', 'September 2021', 'U/ml', 'Mrs. Anderson'."
49         )
50         job_manifest = JobManifest(
51             chunk_id=f"doc_id_chunk_id",
52             task_id=task_id,
53             chunk=chunk,
54             task=task,
55             advice="Focus on extracting the specific keywords related to Mrs. Anderson's tumor marker
      levels."
56         )
57         job_manifests.append(job_manifest)
58 ```
```

**P_worker**

```
 1 Your job is to complete the following task using only the context below. The context is a chunk of text
       taken arbitrarily from a document, it might or might not contain relevant information to the task.
 2
 3 ## Document
 4 {context}
 5
 6 ### Question you are trying to answer:
 7 {question}
 8
 9 # You have been instructed to extract information pertaining to the following concepts:
10 # \"Date of visit\", {task}
11
12 Format your response as follows:
13 {{
14 "Date of visit" : "`direct quote extracted text`",
15 "<keyword_1>" : "`direct quote extracted text`",
16 "<keyword_2>" : "`direct quote extracted text`",
17 ...
18 }}
19
20 Can you please extract the relevant sections from the document that are related to the concepts
       provided? Extract direct quotes or sentences. If concept is not mentioned, leave it out.
21
22 Your Answer:
```

**P**synthesize

```
 1 Answer the following by the synthesizing findings from multiple junior workers (LLMs).
 2
 3
 4 ---
 5 ## Inputs
 6 1. Question to answer:
 7 {question}
 8
 9 2. Collected Job Outputs (from junior models):
10 {extractions}
11
12 ---
13 First think step-by-step and then answer the question using the exact format below.
14
15 ## ANSWER GUIDELINES
16
17 **Required JSON Output**: You must output exactly one JSON object with these keys:
18     - "decision": Must be "provide_final_answer" or "need more information"
19     - "explanation": A short statement about how you arrived at your conclusion or what is still missing
       .
20     - "answer": a final answer that is a text span pulled directly from the job output citations.
21
22
23 Here is the template for your JSON response:
24
25 <think step-by-step here>
26
27 {{
28 "decision": "...",
29 "explanation": "...",
30 "answer": "..,",
31 }}
32
33 Now, carefully inspect the question, think step-by-step and perform any calculations before outputting
       the JSON object.
34 - If answer choices are provided, your answer must **exactly** match one of the answer choices.
35 - Don't paraphrase the final answer --- extract text directly from the document(s) or previous job
       outputs.
36
37 Question:
38 {question}
39
40 Your Answer:
```

# G. Generated code

## G.1. Minions: Example generated code for task decomposition

The document is a 10k statement for Amazon. The question is: *"What is the breakdown of Amazon's operating expenses as a percentage of net sales for 2019?"*. The code generated by the remote model for creating task specification (first round) appears below. The code includes to functions, one to prepare the list of job specifications, and one to aggregate the outputs returned from executing these jobs, formatting them into a single string.

```python
1  def prepare_jobs(
2      context: List[str],
3      prev_job_manifests: Optional[List[JobManifest]] = None,
4      prev_job_outputs: Optional[List[JobOutput]] = None,
5  ) -> List[JobManifest]:
6      # Chunk the document into manageable sections
7      chunks = chunk_by_section(context[0], max_chunk_size=3000, overlap=20)
8
9      # Define the task to extract operating expenses and net sales
10     task = "Extract the operating expenses and net sales figures for 2019 from this chunk. Include any
        percentages if available."
11
12     # Create job manifests for each chunk
13     job_manifests = []
14     for chunk in chunks:
15         job_manifests.append(JobManifest(chunk=chunk, task=task, advice="Focus on financial data and
        percentages."))
16
17     return job_manifests
18
19 def transform_outputs(
20     jobs: List[Job],
21 ) -> str:
22     # Initialize a dictionary to store extracted data
23     extracted_data = {}
24
25     # Process each job output
26     for job in jobs:
27         output = job.output
28         if output.answer:
29             # Aggregate the answers by chunk_id
30             chunk_id = job.manifest.chunk_id
31             if chunk_id not in extracted_data:
32                 extracted_data[chunk_id] = []
33             extracted_data[chunk_id].append(output.answer)
34
35     # Combine all extracted data into a single string for further analysis
36     aggregated_output = ""
37     for chunk_id, answers in extracted_data.items():
38         aggregated_output += f"Chunk {chunk_id}:\n"
39         for answer in answers:
40             aggregated_output += f"{answer}\n"
41
42     return aggregated_output
```

For this example query, the model decides to chunk the context with a maximum of 3000 characters per chunk (line 7). It then assigns a single extraction task to be performed on all chunks (line 10).

Note that a generic character-based chunking function is specified by us and made available in the global workspace during code execution:

```python
1  def chunk_by_section(
2      doc: str, max_chunk_size: int = 3000, overlap: int = 20
3  ) -> List[str]:
4      sections = []
5      start = 0
6      while start < len(doc):
7          end = start + max_chunk_size
8          sections.append(doc[start:end])
9          start += max_chunk_size - overlap
10     return sections
```

Our framework enables providing a library of chunking functions to select from. Future work should have the model come up with the chunking function itself.

## G.2. Improving Minions with Supervised Finetuning

*Can we make a better minion through finetuning?* There are at least two possibilities for doing so:

- **Training on the domain.** Having the small model specialize on the target domain, by training it to solve problems and answer questions on input documents from the new domain.

- **Training to be a Minion.** Training the small model to produce correct and concise responses to a supervisor model, to arrive at correct responses quickly.

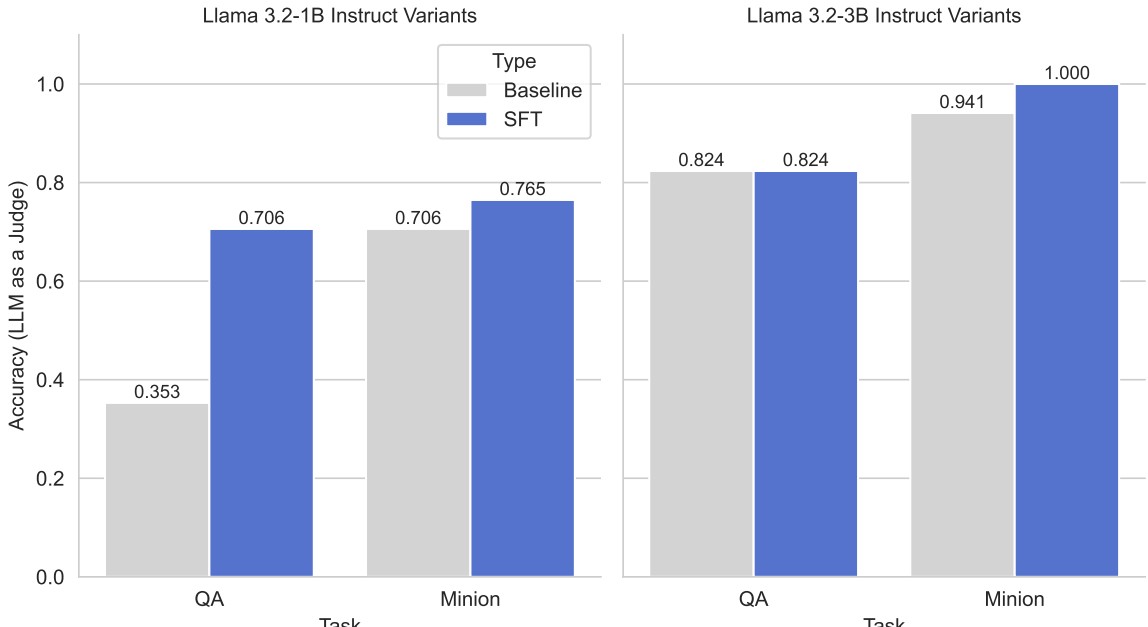

*Figure 11.* **Supervised finetuning improves minion performance on LongHealth.** We compare baseline QA (answering a query on a document) and MINION protocols, with and without finetuning (blue and gray, respectively), using LLM-as-a-judge accuracy (y-axis). Left: 1B LocalLM. Right: 3B LocalLM.

Here we present *preliminary* results on the first possibility.

**Experimental Setup** We finetune small models (LLAMA-3.2-1B-Instruct and LLAMA-3.2-3B-Instruct) on the LONGHEALTH dataset (Adams et al., 2024). We train on 317 questions and test on 17 held-out questions. We finetune models using Low-Rank Adaptation (LoRA; (Hu et al., 2022)), following best practices for hyperparameter selection ($r = 16$, $\alpha = 2r$, dropout0.05, targeting all modules of the transformer and training for four epochs; (Biderman et al., 2024)), and sweeping over learning rates (best LR for 1B was $1e - 4$ and 3B $2e - 4$).

**Evaluation** We use LLM-as-a-Judge (Zheng et al., 2023) with GPT-4O to determine if the predicted response matches the ground-truth response, avoiding strict string matching.

For each model size, we analyze accuracy in four conditions:

- Baseline QA – asking the small model to answer a test question the context, without any finetuning or local-remote protocols

- SFT QA – asking a finetuned small model to answer a test question on the context

- Baseline Minion – without any finetuning, using the small model as part of the MINION protocol

- SFT Minion – using a finetuned model as part of MINION

**Results** See Figure 11. Across dataset regimes, we find that Minion is superior to vanilla QA, that is, the small model does better when it is guided by a frontier model that asks one question at a time until reaching a solution. For 1B but not 3B, SFT improves the baseline QA performance. Using the SFT'd checkpoints as part of the Minion protocol results in further gains for both model sizes.

We caveat that this is a relatively small finetuning dataset, and that future work is needed to reproduce the results on larger scales.

