# OpenReview forum: "Cost-efficient Collaboration between On-device and Cloud Language Models"
_ICML.cc/2025/Conference — ICML 2025 poster_

### Official Review · Reviewer_5xc4 · 2025-03-07

**Overall Recommendation:** 4

**Summary:**

The paper presents a setting where a small model having access to local data collaborates with a state-of-the-art LLM cloud-hosted (without access to the data) to solve real tasks. To improve over an initial naive protocol (with back and forth chats between the two models), the paper introduces Minions, where the cloud model creates sub-tasks for the local model to execute. The paper presents clear reduction in costs while maintaining high performances.

**Claims And Evidence:**

The core of the paper is focused around presenting processes that reduce cost while maintaining high performance (see for instance Figure 2). Overall the claims of cost reduction while maintaining high performance are convincing and well supported by evidence (e.g. Fig 5-6 and Table 1) and overall the paper is very well structured making it a strong contribution to the conference.

**Essential References Not Discussed:**

I haven't noticed anything missing

**Experimental Designs Or Analyses:**

The experimental design is the core strength of this paper: it is well planned and shows the advantage of Minions over the initial naive protocol and in comparison to the state of the art performance of a frontier model (GPT-4o)

**Methods And Evaluation Criteria:**

The method and evaluation criteria are robust (see Table 1) and especially the benchmark across different type of datasets (finance, health and scientific QA) gives a clear overview of benefits of this approach.

**Other Comments Or Suggestions:**

I would improve Figure 2, which I think it's very important for the narrative of the paper, but I found a bit hard to read as many layers of information are embedded in it

**Other Strengths And Weaknesses:**

The main point which I would like the author to discuss more in details is a setting where the data on premise is sensitive and should absolutely not be shared with the cloud model. How would you guarantee that that the local LLM would not leak, even by mistake, any piece of information to the cloud model? Have you done any experiment on this specific topic?

**Questions For Authors:**

Nothing else

**Relation To Broader Scientific Literature:**

The paper is well positioned in the scientific literature (however this is mostly presented in the appendix and not directly in the main content of the paper)

**Theoretical Claims:**

The paper doesn't rely on theoretical claims

---

> ### Author Rebuttal · Authors · 2025-03-31
>
> Thank you for the detailed feedback! We include a Common Response, followed by an Individual Response.
>
> Please see the revised paper at this anonymous link: https://storage.googleapis.com/anonymous-files/minions.pdf
>
> ## Common Response
>
> We appreciate the positive feedback from all the reviewers:
>
> - Local-remote systems are a “compelling” \[xykf] and “underexplored direction of research” \[vVmz], making our submission a “strong contribution to the conference” \[5xc4].
>
> - Our central claim – that such systems offer attractive cost-accuracy tradeoffs – is substantiated \[xykf], “well-supported by evidence” \[5xc4], and “backed by experiments” \[vVmz].
>
> - The experimental setup is “thorough” \[vVmz], “well-planned” \[5xc4], and “comprehensive” \[xykf], and forms the “core strength of the paper” \[5xc4].
>
> The reviewers’ feedback motivated the following updates:
>
> - **Latency \[vVmz,xykf]:** We added latency benchmarks on a single consumer GPU (_e.g._ RTX 4090) and found Minion/MinionS are only 1.44× and 2.89× slower than remote-only, while yielding 30× and \~5× cost savings (See §6.5)
>
> * **Adapting LocalLM \[vVmz]:** We show that Minion accuracy can be improved by supervised finetuning (SFT) of the minion (1B & 3B scales) on the target domain (§G.2). In §7, we highlight further opportunities for co-adaptation.
>
> * **Agentic Tool Use \[vVmz]:** Introduced a tool-augmented version of Minions, where the local model can use local tools, matching GPT4o-only accuracy (0.7) while cutting prefill cost by \~3.7× (See §E.5)
>
> * **Energy savings \[xykf]:** Added energy consumption analysis showing Minions consumes just 1/12th (1B LocalLM) and 1/6th (3B LocalLM) the energy of GPT-4o alone (see §E.4).
>
> * **Privacy implications \[xykf,5xc4]:** We highlight this important direction in §7. While privacy merits a more careful treatment in future work, our preliminary results show that local LLM-based PII filtering reduces PII leakage from 22% to 4.5%
>
> ---
>
> ## Individual Response
> **Exploration on sensitive data.**
>
> > _“The main point which I would like the author to discuss more in details is a setting where the data on premise is sensitive and should absolutely not be shared with the cloud model. How would you guarantee that that the local LLM would not leak, even by mistake, any piece of information to the cloud model? Have you done any experiment on this specific topic?_
>
> Thank you for highlighting this important setting! While a comprehensive treatment of data privacy  is beyond the scope of this paper, we agree that the topic merits a preliminary exploration. We thus present an initial analysis of data leakage mitigation within the Minions framework using  contemporary privacy-preserving techniques, including controlled prompting [1,2] and PII filtering. Methodologically, we use a prompt-based filtering layer—a secondary LLM call applied to local LMs output—to remove sensitive information. We evaluate this method on a QA dataset over a local filesystem of purchase receipts containing emails, addresses, phone numbers, credit card details, and more. With the privacy filter enabled, PII leakage rate drops from 22% to 4.5%. While preliminary and imperfect, these results highlight the importance of further research in this area.
>
> **_Clarity and Presentation._**
>
> As per your feedback, we have improved the clarity of Figure 2.
>
> **\[1]** [**https://arxiv.org/abs/2410.17127**](https://arxiv.org/abs/2410.17127)
>
> **\[2]** [**https://arxiv.org/pdf/2403.03129**](https://arxiv.org/pdf/2403.03129)

---

> > ### Comment · Reviewer_5xc4 · 2025-04-02
> >
> > Thank you for this, I'm happy with your response which addressed my main point

---

> > > ### Author Response · Authors · 2025-04-06
> > >
> > > Thank you for taking the time to review our work and for highlighting the importance of settings with sensitive data. Your suggestions improved our manuscript!

---

### Official Review · Reviewer_xykf · 2025-03-12

**Overall Recommendation:** 3

**Summary:**

This paper presents MINION and MINIONS, novel frameworks for cost-efficient collaboration between small on-device and cloud-based language models. MINION enables asymmetric collaborative communication between LocalLM (Reading) and RemoteLM (Reasoning), achieving a 30.4× cost reduction while recovering 87% of remote-only performance. However, it struggles with multi-step instructions and long contexts. To address this, MINIONS introduces task decomposition and parallelized subtasks in LocalLM, reducing costs by 5.7× while recovering 97.9% of remote-only performance. Experimental validation on multiple benchmarks highlights the trade-off between cost and accuracy.

**Claims And Evidence:**

This paper asserts that the MINIONS protocol significantly reduces cloud inference costs while maintaining accuracy comparable to cloud-based models. Additionally, it claims that by adjusting the protocol’s hyperparameters, a flexible trade-off between cost and performance can be achieved. To support this claim, the authors conduct extensive evaluations across various domain benchmarks and different model sizes and types, reinforcing the validity of their argument. Notably, the analysis of performance recovery based on model size substantiates the claim regarding the cost-performance trade-off.

**Essential References Not Discussed:**

This paper leverages the appendix to cite all relevant studies comprehensively.

**Experimental Designs Or Analyses:**

The experimental design and analysis are conducted using appropriate methodologies. However, there is a lack of information on the experimental setup, such as the specific GPU used, which should be supplemented. Additionally, the comparative analysis of different hyperparameters effectively identifies the key factors influencing the protocol’s performance.

**Methods And Evaluation Criteria:**

The proposed MINIONS framework is well-suited to the problem and highly plausible. The explicit role definition for both language models and the protocol designed to compensate for the limitations of small LMs are particularly compelling. However, the explanation of how the method operates in each iteration is located in the appendix, making it difficult to find and understand the exact workings of the loop. It would be beneficial to include a brief explanation in the main text or provide a hyperlink to the appendix. The evaluation criteria focus on accuracy and cost, both of which are highly appropriate for assessing the proposed protocol’s cost-performance trade-off.

**Other Comments Or Suggestions:**

- It might be useful to explore privacy-aware chunk extraction to enhance secure collaboration between LocalLM and RemoteLM.
- Maintaining consistent terminology (e.g., ensuring "multi-step" and "multi-part" are clearly defined) would help avoid potential confusion.
- In line 246, clarifying whether "cloud model" refers to RemoteLM in this study would improve clarity.

**Other Strengths And Weaknesses:**

Strength
- The multi-round communication approach introduced in the framework allows for iterative improvements in performance, which is a promising direction.
- The paper provides an insightful cost analysis, demonstrating how task decomposition and parallelization contribute to cost reduction. Also, the study does well in quantifying the trade-offs between cost and accuracy across multiple benchmarks.
- The inclusion of various hyperparameter evaluations strengthens the reliability of the findings.
- Sharing prompts used in experiments, along with example responses, enhances reproducibility and improves clarity.

Weakness
- The size of Local LM might be too large for practical on-device deployment in resource-constrained environments. Discussing the trade-offs between model size and performance in more detail would provide valuable insights into the feasibility of different approaches.
- While cost reductions are well-analyzed, a more detailed discussion of latency optimization would be helpful. Measuring actual latency could provide clearer performance.
- The paper does not extensively discuss energy consumption or the impact on local resources (e.g., memory and computational overhead). A deeper analysis of these aspects would be valuable.
- The framework would benefit from a clearer pipeline structure, particularly in explaining how the loop functions, as described in Section D. The current explanation is somewhat unclear, making it difficult to fully understand the method and integrate it into existing workflows. Additionally, providing example data in the appendix to illustrate how the Remote LM applies its data chunking strategy would be helpful. The inconsistent use of terminology also makes the paper harder to follow on the first read. If these aspects were clarified and the protocol were made more explicit, I would be willing to reconsider my rating.

**Questions For Authors:**

- Does the framework remain effective for small language models under 1B parameters?
- Has the impact of network latency on local-remote communication performance been analyzed?
- Are there potential security risks, such as adversarial attacks, in local-remote collaboration?
- Can this approach be extended to other multimodal inputs?

**Relation To Broader Scientific Literature:**

This study builds upon prior research in multi-agent systems, retrieval-augmented generation (RAG), and cost-efficient LLM routing. Notably, it differentiates itself by addressing asymmetric roles between small on-device models and large cloud-based models, as well as exploring inter-model interactions. Citations to relevant literature are sufficient, and comparative experiments with RAG models effectively demonstrate the value of MINIONS.

**Theoretical Claims:**

Rather than relying on theoretical claims, this study is primarily supported by experimental evidence, which is appropriate for the research context. However, a comparative analysis of communication latency between the two language models alongside real experimental data would have enhanced credibility.

---

> ### Author Rebuttal · Authors · 2025-03-31
>
> Thank you for the detailed feedback! We include a Common Response, followed by an Individual Response.
>
> Please see the revised paper at this anonymous link: https://storage.googleapis.com/anonymous-files/minions.pdf
>
> ## Common Response
>
> We appreciate the positive feedback from all the reviewers:
>
> - Local-remote systems are a “compelling” \[xykf] and “underexplored direction of research” \[vVmz], making our submission a “strong contribution to the conference” \[5xc4].
>
> - Our central claim – that such systems offer attractive cost-accuracy tradeoffs – is substantiated \[xykf], “well-supported by evidence” \[5xc4], and “backed by experiments” \[vVmz].
>
> - The experimental setup is “thorough” \[vVmz], “well-planned” \[5xc4], and “comprehensive” \[xykf], and forms the “core strength of the paper” \[5xc4].
>
> The reviewers’ feedback motivated the following updates:
>
> - **Latency \[vVmz,xykf]:** We added latency benchmarks on a single consumer GPU (_e.g._ RTX 4090) and found Minion/MinionS are only 1.44× and 2.89× slower than remote-only, while yielding 30× and \~5× cost savings (See §6.5)
>
> * **Adapting LocalLM \[vVmz]:** We show that Minion accuracy can be improved by supervised finetuning (SFT) of the minion (1B & 3B scales) on the target domain (§G.2). In §7, we highlight further opportunities for co-adaptation.
>
> * **Agentic Tool Use \[vVmz]:** Introduced a tool-augmented version of Minions, where the local model can use local tools, matching GPT4o-only accuracy (0.7) while cutting prefill cost by \~3.7× (See §E.5)
>
> * **Energy savings \[xykf]:** Added energy consumption analysis showing Minions consumes just 1/12th (1B LocalLM) and 1/6th (3B LocalLM) the energy of GPT-4o alone (see §E.4).
>
> * **Privacy implications \[xykf,5xc4]:** We highlight this important direction in §7. While privacy merits a more careful treatment in future work, our preliminary results show that local LLM-based PII filtering reduces PII leakage from 22% to 4.5%
>
> ---
>
> ## Individual Response
>
> **Latency Analysis**
>
> > A more detailed discussion of latency optimization would be helpful.
>
> The updated §6 includes comprehensive latency experiments. See §6, the Common Response and the individual response to vVmz for more details.
>
> **Feasibility**
>
> > The size of Local LM might be too large for practical on-device deployment.
>
> §6.2 now discusses the feasibility of running LLMS on modern laptops and workstations¹. Devices like the d MacBook Pro support up to 600B and 200B models with quantization², and even iPhone 15 Pro handles \~3B³. Our latency experiments on consumer grade hardware show Minion and Minions are only 1.44× and 2.89× slower than remote-only (see §6).
>
> \[1] <https://ollama.com/library>
>
> \[2] <https://www.apple.com/newsroom/2024/10>
>
> \[3] <https://machinelearning.apple.com/research/introducing-apple-foundation-models>
>
> **Model size tradeoffs**
>
> > Discussing the trade-offs between model size and performance in more detail would provide valuable insights into the feasibility.
>
> In our revised manuscript, §6.2 (Model Choice) and Figure 4 detail how performance and communication efficiency vary with local model size.
>
> **Energy consumption analysis**
>
> > The paper does not extensively discuss energy consumption.
>
> Your point led to a new analysis, showing major energy savings by Minions (see  §E.4 for analysis). As we do not have access to the hardware running GPT-4o, we use energy consumption estimates from Epoch AI [1].  We benchmark local energy use for 1B & 3B models on an M1 Max and A100 GPU. We compare to GPT-4o-only execution, and find 12$\times$ energy savings with the 1B LM and 6$\times$ with the 3B model on A100, with similar gains on M1 Max.
>
> \[1] https\://epoch.ai/gradient-updates/how-much-energy-does-chatgpt-use
>
> > Might be useful to explore privacy-aware chunk extraction to enhance secure collaboration.
>
> We agree! While a full treatment of data privacy is beyond the scope of this paper, we report preliminary experiments on leakage mitigation within Minions (see Common Response and response to 5xc4).
>
> **Clarity and Presentation**
>
> - Experimental Setup: We added hardware details for local models (see §B).
>
> - Protocol Description: We rewrote the Methods accordingly; added an example of remote chunking (see §4,5,G.1).
>
> - Consistent Terminology
>
>   - The revision uses a more consistent terminology for the names of the local and remote models.
>
>   - Consolidated "multi-step" and "multi-part" language.
>
> **Other Questions**
>
> > Does this work with small models (<1B)?
>
> Not well—performance improves significantly with local model sizes >=3B (see §6 + Fig. 4)
>
> > Is network latency a bottleneck?
>
> No, communication time is negligible (<0.002%) compared to inference (see §E.6)
>
> > Are there security concerns?
>
> Yes! There are works that study this extensively [1].
>
> > Can this support multimodal inputs?
>
> Yes, using a VLM as the local LM enables image-text processing (see §E.7)
>
> \[1] <https://arxiv.org/abs/2310.06845>

---

> > ### Comment · Reviewer_xykf · 2025-04-04
> >
> > The authors clearly addressed my concerns regarding latency, on-device feasibility, and energy consumption, with the revised version effectively highlighting the framework’s strength—most notably, 12× reduction in energy consumption. Also the revised version provided a clearer understanding of the overall framework operation, which motivated an upward adjustment in the rating. I raised the scoring to 3.

---

> > > ### Author Response · Authors · 2025-04-06
> > >
> > > Thank you for taking the time to review our work and for the valuable feedback around cost experiments and presentation clarity which has improved the manuscript!

---

### Official Review · Reviewer_vVmz · 2025-03-12

**Overall Recommendation:** 4

**Summary:**

The paper proposes an agentic pattern for collaborative modeling between a cloud-based large LM and a client-side small LM to reduce cloud inference costs. The authors propose two approaches:
1. MINION: A simple communication protocol where the small model summarizes and interacts with the cloud model. However, it struggles with long contexts and following multi-step instructions.
2. MINIONS: An improved systems approach that decomposes tasks into smaller chunks, enabling more efficient execution and parallelization.

The authors show that MINIONS recovers 97.9% of the accuracy of a cloud-only model while reducing costs by 5.7×. They also discuss ways to optimize the client LM, such as parallelizing subtasks to improve efficiency.
The study evaluates a range of client models and finds that this approach is effective for models above 3B parameters, with performance improving as model size increases.

**Claims And Evidence:**

Claims are reasonably well supported and backed by experiments.

**Essential References Not Discussed:**

There is other work leveraging a smaller model combined with a big one. Consider citing the following:
Prompt compression using a smaller model. This will reduce cloud efficiency and potentially also could run on client
Jiang et al. (2023). LLMLingua: Compressing Prompts for Accelerated Inference of Large Language Models. EMNLP 2023.
Pan et al. (2024). LLMLingua-2: Data Distillation for Efficient and Faithful Task-Agnostic Prompt Compression. ACL 2024.

Consider referencing for collaborative cloud/edge modeling (and related references):
Hao et al (2024) Hybrid SLM and LLM for Edge-Cloud Collaborative Inference. EdgeFM 2024
Xia et al. (2024). Hybrid Retrieval-Augmented Generation for Real-time Composition Assistance, EMNLP Industry Track 2024

**Experimental Designs Or Analyses:**

The authors perform a comprehensive set of experiments. Their setup and analysis seems sound, but there is a definitve gap in evaluating latency as pointed out above.

**Methods And Evaluation Criteria:**

The selected datasets cover a diverse set of domains domains in finance, healthcare, and science. It would have been desirable to include and discuss evaluation methods that more explicitly span tasks of varying complexity (e..g. simple Q&A retrieval tasks, more complex reasoning, ...). Also, given the setup for cloud/client, agentic tasks such as automatically perform certain actions would have been relevant to include.

Lastly, the main evaluation is cost and performance. One major reason for client side inference is closeness to user, and providing a snappy experience. Beside a little mention at the beginning, there is no evaluation of latency across the paper.I would expect that this approach would significantly increase latency and diminish value of this approach.

**Other Comments Or Suggestions:**

n/a

**Other Strengths And Weaknesses:**

Strengths:
- Interesting take on cloud/client models of agent scenarios
- thorough experimental study (despite lack broader benchmarks)

Weakness:
- Mainly experimental with little theoretical backing
- Main ML aspect is prompt engineering, and combination of multiple agents. Adapting (or at least the discussion of it) the small model to this interaction pattern would be an interesting ml aspect and extension.

**Questions For Authors:**

n/a

**Relation To Broader Scientific Literature:**

The paper makes a contribution in the very crowded space of LLM agents. It's angle on splitting agents across cloud and client is generally an underexplored area and interesting direction of research.

The papers main positioning seems to be reducing cloud inferencing cost. As such, a more detailed comparison with alternative approaches such as prompt compression, speculative decoding may be desirable. Alternatively, the author should double down more on the benefits of cloud/client setup, and what type of experiences this could enable.

**Theoretical Claims:**

I did not check correctness of theoretical claims

---

> ### Author Rebuttal · Authors · 2025-03-31
>
> Thank you for the detailed feedback! We include a Common Response, followed by an Individual Response.
>
> Please see the revised paper at this anonymous link: https://storage.googleapis.com/anonymous-files/minions.pdf
>
> ## Common Response
>
> We appreciate the positive feedback from all the reviewers:
>
> - Local-remote systems are a “compelling” \[xykf] and “underexplored direction of research” \[vVmz], making our submission a “strong contribution to the conference” \[5xc4].
>
> - Our central claim – that such systems offer attractive cost-accuracy tradeoffs – is substantiated \[xykf], “well-supported by evidence” \[5xc4], and “backed by experiments” \[vVmz].
>
> - The experimental setup is “thorough” \[vVmz], “well-planned” \[5xc4], and “comprehensive” \[xykf], and forms the “core strength of the paper” \[5xc4].
>
> The reviewers’ feedback motivated the following updates:
>
> - **Latency \[vVmz,xykf]:** We added latency benchmarks on a single consumer GPU (_e.g._ RTX 4090) and found Minion/MinionS are only 1.44× and 2.89× slower than remote-only, while yielding 30× and \~5× cost savings (See §6.5)
>
> * **Adapting LocalLM \[vVmz]:** We show that Minion accuracy can be improved by supervised finetuning (SFT) of the minion (1B & 3B scales) on the target domain (§G.2). In §7, we highlight further opportunities for co-adaptation.
>
> * **Agentic Tool Use \[vVmz]:** Introduced a tool-augmented version of Minions, where the local model can use local tools, matching GPT4o-only accuracy (0.7) while cutting prefill cost by \~3.7× (See §E.5)
>
> * **Energy savings \[xykf]:** Added energy consumption analysis showing Minions consumes just 1/12th (1B LocalLM) and 1/6th (3B LocalLM) the energy of GPT-4o alone (see §E.4).
>
> * **Privacy implications \[xykf,5xc4]:** We highlight this important direction in §7. While privacy merits a more careful treatment in future work, our preliminary results show that local LLM-based PII filtering reduces PII leakage from 22% to 4.5%
>
> ---
>
> ## Individual Response
>
> **Latency analysis**
>
> > There is no evaluation of latency in the paper. I would expect that this approach would increase latency.
>
> Your suggestion led to new latency experiments to complement the theoretical analysis in the original submission. We benchmarked Minion/MinionS on 2 consumer-grade GPUs commonly used in local workstations (e.g., RTX 4090, MSRP 1,599 USD on 03/26), finding they are only $1.44×$ and $2.89×$ slower than remote-only, while offering \~30× and \~5× cost savings. See Tab. 2 in the revised manuscript for more details.
>
> We note that these empirical latency measurements depend on point-in-time factors like local hardware and cloud load. Thus, in §C.2 we provide a theoretical framework for estimating the latency overhead of any local-remote system given model and hardware specifications like memory bandwidth.
>
> **Minion + agentic tool use.**
>
> > Given the setup for cloud/client, agentic tasks such as automatically perform certain actions would have been relevant to include.
>
> To support agentic tasks—where the model autonomously performs actions—we extend the Minions framework to enable local tool use (see §E.5). In this setup, the local LM executes actions guided by the remote LM. We evaluate this on filesystem queries with 5 tools and find that using Qwen2.5 locally with GPT-4o as the remote LM, matches the performance of GPT-4o-only while using less than 28% of the remote tokens (see Tab. 12).
>
> **Expansion of related works.**
>
> We now cite several of the works you highlighted including speculative decoding, collaborative inference, and prompt compression.
>
> **Adapting LocalLM**
>
> > Adapting the small model to this interaction pattern would be an interesting ml aspect and extension.
>
> See common response. The revised manuscript now takes a step in this direction by finetuning small models on the target domain and demonstrating improvements in Minion accuracy (§G.2). The revised discussion (see “Local-remote model co-design”) spells out a number of extensions we are excited about, including multi-agent co-training.
>
> **Stratifying by task complexity**
>
> > It would have been desirable to include evaluations that more explicitly span tasks of varying complexity
>
> We agree and therefore stratify Financebench (FIN) and Longhealth (LH) problems by complexity. Surprisingly, we find that the Minions protocol outperforms the remote-only condition on harder problems. For example, on simple info. extraction tasks in FIN, Minions (with Qwen2.5) trails by 22.7 points, but on complex extraction & numerical reasoning tasks, it outperforms the remote-only by +4.6. Same holds for LH, where Minions (with Llama-8B) is -6.2 pts. on single-span questions but leads by +16.0 on multi-span synthesis. This trend holds across model sizes.
>
> **Expanded discussion of cloud/client setup**
>
> > Alternatively, the author should double down more on the benefits of cloud/client setup…
>
> We’ve expanded §7 (Discussion) to better highlight the benefits of the cloud/client setup.

---

### Decision · Program_Chairs · 2025-05-01

**Decision:**

Accept (poster)

**Comment:**

The paper proposes a framework designed to enable efficient collaboration between a small client-side language model and a large cloud-based model, with the primary goal of significantly reducing cloud inference costs while maintaining competitive performance. All reviewers appreciate the paper’s novel approach and rigorous experimental validation, noting its relevance in the rapidly evolving domain of collaborative LLM agents. Strengths highlighted include comprehensive experimentation across diverse domains, clearly demonstrated cost-performance trade-offs, insightful analyses on hyperparameter tuning, and robust supplementary materials for reproducibility. The reviewers find the paper's claims well-supported by extensive empirical evidence.

The authors and reviewers have a good rebuttal discussion. The authors are encouraged to incorporate these discussion and findings in the camera ready revision.